# Precisely Targeted Nanoparticles for CRISPR-Cas9 Delivery in Clinical Applications

**DOI:** 10.3390/nano15070540

**Published:** 2025-04-02

**Authors:** Xinmei Liu, Mengyu Gao, Ji Bao

**Affiliations:** Department of Pathology, Institute of Clinical Pathology, Key Laboratory of Transplant Engineering and Immunology, National Health Commission of China, West China Hospital, Sichuan University, Chengdu 610041, China

**Keywords:** CRISPR-Cas9, genome editing, nanoparticles, applications

## Abstract

Clustered regularly interspaced short palindromic repeats/CRISPR-associated protein 9 (CRISPR-Cas9), an emerging gene-editing technology, has recently gained rapidly increasing attention. However, the lack of efficient delivery vectors to deliver CRISPR-Cas9 to specific cells or tissues has hindered the translation of this biotechnology into clinical applications. Chemically synthesized nanoparticles (NPs), as attractive non-viral delivery platforms for CRISPR-Cas9, have been extensively investigated because of their unique characteristics, such as controllable size, high stability, multi-functionality, bio-responsive behavior, biocompatibility, and versatility in chemistry. In this review, the key considerations for the precise design of chemically synthesized-based nanoparticles include efficient encapsulation, cellular uptake, the targeting of specific tissues and cells, endosomal escape, and controlled release. We discuss cutting-edge strategies to integrate chemical modifications into non-viral nanoparticles that guide the CRISPR-Cas9 genome-editing machinery to specific edits. We also highlighted the rationale of intelligent nanoparticle design. In particular, we have summarized promising functional groups and molecules that can effectively optimize carrier function. In addition, this review focuses on advances in the widespread application of NPs delivery in the biomedical fields to promote the development of safe, specific, and efficient NPs for delivering CRISPR-Cas9 systems, providing references for accelerating their clinical translational applications.

## 1. Introduction

With the development of biotechnology, the successive exploration of zinc finger nucleases (ZFNs) and transcription activator-like effector nucleases (TALENs) have greatly promoted the development of gene-editing technology [1,2,3]. As a third-generation gene-editing technology, clustered regularly interspaced short palindromic repeats/CRISPR-associated protein 9 (CRISPR-Cas9) is more flexible, effective, and precise and thus has more application potential [4,5]. There are typically three forms of delivery of the CRISPR-Cas9 system, which include DNA, messenger RNA (mRNA), and ribonucleoprotein (RNP), and all three different forms have their own advantages and limitations. So, the selection of these forms usually relies on the selection of delivery vectors and downstream applications [4,6]. Although the CRISPR-Cas9 system is more powerful in genome editing than ZFNs and TALENs, several practical issues and technical challenges must be addressed before use in the clinic [7].

Efficient delivery of the CRISPR-Cas9 system, which is essential for efficient editing of genes and their clinical applications, has received increasing attention in recent years [8]. The existing delivery methods include physical methods, viral vector-based delivery, and non-viral vector-based delivery. Physical delivery methods (electroporation, microinjection, and micro-fluidics) could deliver CRISPR systems directly into cells with high transfection efficiency; however, a specialized device is required and is generally not suitable for in vivo delivery [6,9,10]. Adenovirus (AV), adeno-associated virus (AAV), lentiviral vectors (LVs), and other viral vectors are widely used in CRISPR-Cas9 system delivery to achieve efficient transfection efficiency and stable transgene expression, but their immunogenicity and mutational risks are still not negligible [5,11,12]. The above limitations have accelerated the development of non-viral vectors. Through the reasonable design of non-viral vectors, the above limitations could be overcome to expand the application scope of CRISPR-Cas9 systems. However, in vivo delivery of the CRISPR-Cas9-based genome editor through viral or non-viral vectors is inevitably influenced by non-specific distribution upon intravenous administration, leading to accumulation in non-targeted organs and tissues. Therefore, unnecessary editing may result in unpredictable genetic toxicity and serious side effects [13]. At present, there is an urgent need to explore suitable delivery tools and engineering methods to achieve more efficient and accurate delivery of CRISPR-Cas9 for in vivo gene editing.

With the advancement of nanotechnology, a wide range of non-viral delivery systems for genome editing have been developed. Nanotechnology is widely used for delivering gene therapy agents, including antisense oligonucleotides, siRNAs, microRNAs, mRNAs, and CRISPR-Cas9 [14]. Recent research results suggest the use of synthetic nanoparticles as versatile CRISPR-Cas9-delivery tools that could be adapted for experimentally studying the biology of cancer as well as for clinically translating cancer gene therapy for the following key advantages. First, the Cas9 protein and single guide RNA (sgRNA) can be co-encapsulated even when they are large in size; another appealing feature of nanoparticles is that their geometrical conformation and surface characteristics could be modified in favor of the cargos released at targeted sites to avoid side effects. In addition, targeting specific tissues or organs can also be achieved by adjusting the composition, size, shape, and surface properties of nanoparticles. Last but not least, chemical synthesis meets the needs of stable large-scale production. So far, a variety of nanocarriers, including synthetic organic nanoparticles based on liposomes [15,16,17,18] and polymers [19,20,21], inorganic nanoparticles such as gold nanoparticles [22,23,24,25,26,27,28,29], nano-silicon dioxide [30,31,32,33,34,35], nano-iron oxide [36,37,38], etc., as well as native extracellular vesicles [39,40,41,42], with different delivery approaches have been developed for various therapeutic modalities. The classification of such nanoparticles is shown in Figure 1.

As promising gene delivery vectors, nanoparticles still face critical challenges in enhancing therapeutic efficacy and avoiding off-target effects. These issues, especially the uncontrolled release and low transfection efficiency, remain to be further addressed. This review summarizes the advantages and commonalities of nanoparticle approaches previously documented and synthesizes cutting-edge advances in nanoparticle-enabled CRISPR delivery, with emphasis on four strategic pillars: (1) enhanced payload encapsulation through electrostatic/protein–ligand interactions; (2) targeted delivery via ligand-receptor binding, biomimetic coatings, and selective organ-targeting (SORT) technology; (3) controlled release mechanisms responsive to biochemical/physical stimuli; (4) applications in the treatment of genetic diseases and cancer. We further analyze preclinical studies to establish structure–activity relationships between NP properties and editing outcomes. Finally, we propose a roadmap addressing manufacturing standardization in in vivo safety profiling and regulatory challenges to accelerate clinical translation. Moreover, challenges and possible strategies to accelerate the development of this technology are proposed, and the current status of NPs used for in vitro and in vivo studies in the last five years is presented in Table 1 and Table 2.

## 2. The Strategies to Improve CRISPR-Cas9 Delivery Efficiency

### 2.1. Improve CRISPR-Cas9 Encapsulation Efficiency

The low editing efficiency of CRISPR-Cas9 is partly related to the low loading and delivery efficiency of the nanoparticles. A key bottleneck is the lack of delivery strategies required to enable efficient encapsulation in formats of DNA, mRNA, and RNP, especially for RNP. Existing NPs are typically efficiently encapsulated via electrostatic interactions, base pairing, and bonding of specific components of nanoparticles with nucleic acids or proteins. Encapsulation efficiency and nucleic acid loading are important parameters for optimized delivery vectors.

#### 2.1.1. Based on Electrostatic Interaction

The encapsulation mechanism based on electrostatic interactions is currently one of the most frequently employed strategies. Because nucleic acid molecules are negatively charged, and since sgRNA is negatively charged, the surface of the RNP is also negatively charged. Thus, positively charged nanoparticles (cationic lipids, polymers, peptides, etc.) have been designed for the encapsulation of plasmids, mRNAs, or RNPs by electrostatic interactions. As a commercial transfection reagent, polyethyleneimine of 25 kDa (PEI 25 kDa) has been proven to have extremely high transfection efficiency [26,84,90]. Therefore, it is also generally considered the “gold standard” for small-sized nucleic acid transfection. Although it has poor performance for the transfection of CRISPR-Cas9-based large plasmids, PEI can be easily complexed with other polymers by incubation at room temperature, and coating the surface of nanocarriers with neutral or negative charges with PEI can effectively improve their encapsulation efficiency and proton escape ability [17]. Chen et al. developed a PEI-based hydrogel and cyclodextrin (CD)-engrafted PEI 25 kDa (PEI-CD) with adamantine (AD)-engrafted PEI 25 kDa (PEI-AD), which were able to noncovalently crosslink to form hydrogel through CD–AD-mediated host–guest interactions. The classical lipid 1,2-dioleoyl-3-trimethylammonium-propane chloride salt (DOTAP) was used as a template, and with the inclusion of the hydrogel, the encapsulation efficiency of Cas9 increased from 6.3% to 62.8% [91].

N,N-Bis(2-hydroxyethyl)-N-methyl-N-(2-cholesteryloxycarbonylaminoethyl)ammonium bromide (BHEM-Chol), a new positively charged cholesteryl lipid, can effectively fuse with the cell membrane [92]. BHEM-Chol has been widely studied to improve the encapsulation efficiency of polymers in recent years. As the dose of BHEM-Chol was increased from 0 mg to 3 mg, the encapsulation efficiency of plasmids improved [93].

Protamine, which contains up to 67% arginine residues, is a highly cationic nuclear protein that possesses cell-penetrating properties and nuclear targeting capabilities. After being capped with cationic protamine, the encapsulation of gold nanoclusters (AuCNs) for CRISPR-Cas9 plasmids was significantly enhanced. Targeting the knock-out of the human papillomavirus (HPV) type 18 (HPV18)-E7 oncogene demonstrated a strong ability to inhibit the proliferation of HeLa cells [25].

Another study focused on Cas9, a highly positively charged protein that possesses a net of 20 positive charges, and its high molecular weight made it difficult to pack into vehicles. They also engineered a series of Cas9 proteins with a glutamate peptide tag (E-tag) at the N-terminus of the *Streptococcus pyogenes* (Sp) Cas9 protein and a variable E-tag (En) length, which provided a patch of local negative charges, presumably enabling interaction with cationic arginine gold nanoparticles (ArgNPs). The delivery system was proven to be suitable for any gene [25].

#### 2.1.2. Based on Base-Pair

Taking advantage of the properties of single-stranded DNA (ssDNA) that can base-pair with the guide portion of the Cas9-bound sgRNA, Sun et al. synthesized a biologically inspired yarn-like DNA nano-clew (DNA NC) by rolling circle amplification as a delivery vehicle, which could be partially complementary to the sgRNA, and loading the Cas9 protein together with the sgRNA, resulting in a higher editing efficacy than the cell-penetrating peptide (CPP)-based vector [94]. Similarly, Han’s team designed a certain amount of thiol-modified DNA fragments (DNA-SH) that are partially complementary to the sgRNA sequence. The DNA-SH fragments were reduced and conjugated to CuS NPs to form CuS-DNA and then assembled with an RNP complex based on the principle of base complementarity, finally mixing them with PEI to obtain CuS-RNP@PEI [86]. Similar strategies have also been designed to encapsulate CRISPR complexes in gold nanoparticles. Gold nanoparticles (GNPs) react with a 5-thiol modified single-stranded oligonucleotide (DNA-SH) that has a region complementary to the donor DNA sequence to form GNP-DNA after being incubated with donor DNA to obtain a GNP-Donor and then mixed with RNP to generate a GNP-Donor-RNP [28].

#### 2.1.3. Based on Protein–Ligand Interactions

There are usually several free functional groups on the surface of proteins, such as imidazole, amine, guanidinium, and anionic carboxylate groups, and another attractive strategy for encapsulating proteins is focused on protein–ligand interactions. As a model cationic polymer, generation 5 (G5) amine-terminated polyamidoamine (PAMAM) dendrimers with well-defined molecular sizes and surface functionalities and PBA-modified dendrimers have been used to deliver 13 cargo proteins with different isoelectric points (pIs) and sizes into the cytosol of living cells, including the Cas9 protein, and could maintain their bioactivity after intracellular release. There are three major types of interactions: nitrogen–boronate complexation, cation–π interactions, and ionic interactions, which enhance the efficiency of polymer delivery of proteins [49]. Similarly, it has been reported that guanidinium-rich lipopeptide-based nanoparticles can significantly increase the RNP loading content (up to 20 wt%) due to guanidinium-rich moieties having a high affinity for the anionic side chains of proteins and the phosphate backbone of sgRNA [51]. Another attractive strategy, the external gold surface of gold nanowires (AuNWs), was modified with 3-mercaptopropionic acid (MPA) to introduce a carboxylic acid surface moiety. Then, MPA-AuNWs were coupled to cysteine via amide bond formation. The cysteine present in the Cas9/sgRNA complex could be immobilized onto the surface of the obtained thiol group functionalized AuNWs through disulfide linkage [23].

#### 2.1.4. Other Influencing Factors

In addition, the shape of nanomaterials is also one factor affecting loading efficiency. In particular, the loading efficiency of gold nanorods (AuNRs) with different aspect ratios (AR) was investigated, and the results revealed that ARPs with higher ARs usually present higher loading capacities. The transfection efficiency becomes optimal when the AR is between 7 and 9, achieving the highest transfection efficiency of up to 79.7% [26]. Cetyltrimethylammonium bromide (CTAB)-mediated synthesis to prepare Au nanorods (ARs) with an aspect ratio of 7.6 was investigated for cancer immunotherapy through synergistic PD-L1 disruption and immunogenic cell death (ICD) activation [44]. The N/P ratio is defined as the amino-to-phosphate ratio, which refers to the ratio of the amino group (N, representing nitrogen) of the nanoparticles (NPs) to the phosphate group (P, representing phosphate) of nucleic acids. The N/P ratio is another important factor influencing encapsulation efficiency. The evaluation of the binding ability of positively charged nanocarriers and nucleic acid generally needs to be carried out by agarose gel electrophoresis. The ability of the hyperbranched cationic structure of HPAA-RGD to bind SGK3-gRNA improves with an increasing N/P ratio. The best binding ability occurs when the N/P ratio is greater than or equal to 10 [50].

### 2.2. Improve CRISPR-Cas9 Delivery Efficiency to the Cytoplasm/Nucleus

While devising new approaches to enable controllable, accurately tuned delivery to the cell nucleus for efficient gene editing, there are three key technological problems worth considering in depth. The efficient cell uptake efficiency, endosome/lysosome escape, and controlled release of the payload into the nucleus (targeting the cell nucleus) have been reported.

#### 2.2.1. Improve the Cellular Uptake Efficiency

Cell membranes are composed of anionic phospholipid bilayers and have a hydrophobic nature, whereas naked nucleic acids are hydrophilic, negatively charged biomolecules, thus exhibiting a limited ability to pass through cell membranes. Endocytosis is an important step in the process by which CRISPR-Cas9 complexes reach their intracellular targets to start gene editing, but it is influenced by many complex factors, including but not limited to size, shape, and surface chemistry. In order to improve the cell uptake efficiency of non-viral vectors, extensive efforts in mechanistic investigations into nano-cell interactions have been made. Understanding the mechanisms by which nanoparticles can be internalized into cells is critical for improving engineered methods. Research indicates that there are two main pathways by which nanomaterials enter cells: direct membrane fusion and endocytosis.

Modifying neutral vectors with positively charged groups is a commonly used strategy to promote membrane fusion. Positively charged nanoparticles (cationic lipids, polymers, peptides, etc.) are promising vectors. However, positively charged nanoparticles are prone to cytotoxicity and immune response, which may hinder their clinical application. In contrast, non-positively charged (negatively charged or neutrally charged) nanoparticles generally exhibit high biocompatibility but are generally difficult for mammalian cells to uptake. Different endocytic pathways have been explored over the years. There is currently a consensus on the five types of endocytosis. Clathrin-coated pit-mediated endocytosis (CME), sometimes referred to as receptor-mediated endocytosis, is the best-understood endocytic route and has benefitted the development of different types of therapeutic strategies [95]. Researchers have reported that some ligands engineered on the nanoparticle surface can induce specific and efficient nanoparticle uptake via receptor-mediated endocytosis mechanisms, such as the folate receptor [19], low-density lipoprotein receptor (LDL-R) [89], and asialoglycoprotein receptor (ASGPr) [96]. Ultrasound (US)-propelled nanomotors are also an attractive strategy to enhance cellular internalization. US-propelled Cas9/sgRNA@AuNWs show a five-fold increase in GFP knock-out after 0.5 h of incubation compared with the passive uptake of static AuNWs under the same conditions [23].

#### 2.2.2. Improve Endosome/Lysosome Escape Efficiency

After cellular uptake, the second obstacle to efficient gene delivery is lysosomal degradation, so nanoparticles with increased endosomal escape abilities are optimal for gene-editor delivery. The proton sponge effect is a predominant mechanism used for endosomal escape. As cationic agents in endosomes capture more protons, the vesicle osmotic pressure changes, facilitating cargo escape from the lysosomal digestive route.

Imidazole groups (pKa~6.0) can be protonated in acidic endocytic compartments. Since the first example of CRISPR-Cas9 delivery by metal-coordination frameworks was reported, studies have shown that ZIF-8 can successfully achieve nuclear delivery of the Cas9 protein and sgRNA with high loading capacity and fast endosomal escape [33]. Despite this, other modified metal–organic coordination NPs have been widely explored for Cas9 delivery. Silica–metal–organic framework hybrid nanoparticles (SMOF NPs) consisting of both the silica and zeolitic imidazole framework (ZIF) were investigated regarding the transfection efficiency in four different cell types, where DNA-loaded and mRNA-loaded SMOF NPs performed with higher transfection efficiency than Lipofectamine 2000 in all cells [31].

#### 2.2.3. Targeting the Cell Nucleus

In the plasmid-based CRISPR-Cas9 system, the final prerequisite for successful genome editing is that CRISPR-Cas9 can be efficiently transported to the cell nucleus for sgRNA and CRISPR transcription through nuclear pore complexes (NPCs). However, trafficking through the cytoplasm to the nucleus is particularly difficult because of the physical diameter of the nuclear envelope barrier, and the upper limit size for transportation into the nucleus by passive diffusion is only 10 nm. PEI and polyethylene glycol (PEG) conjugated carbon quantum dots (CQDs-PP) without any functional modifications were proposed for use in a nuclear targeting CRISPR-Cas9 delivery system. The authors reported that CDQs-PP have a superior nuclear-targeting ability due to their small size, which is sufficient to reach the cell nucleus through NPCs without any functional ligand modification. In addition, compared with the GFP tag, the CQD-PP tag results in more traces of the cellular uptake of the CRISPR-Cas9 plasmid through the green fluorescence signal of the CQDs-PP [48].

Promoting the nuclear uptake path is also a way to improve the efficiency of gene manipulation, which involves the decoration of Cas9 with nuclear targeting molecules and controlling vector size to achieve passive diffusion to the nucleus. Nuclear localization signals (NLS), which act as signal fragments, are generally short peptides that guide the transportation of nuclear proteins from the cytoplasm into the nucleus. The NLS is usually bonded with DNA through electrostatic interactions or inserted into the C-terminus to improve the nuclear targeting of Cas9 [97].

## 3. The Strategies for Targeted Delivery to Specific Tissues or Cells

Nanoparticles deliver CRISPR-Cas9 complexes via systemic or local injection in vivo gene-editing studies and systemically administer CRISPR-Cas9 with a threshold that delivers to specific cells or tissues, leading to unwanted editing. Various approaches have been used to engineer NPs to enhance their targeting ability and therapeutic potency. Here, we reviewed the specific chemistries that have been used to develop targeted gene-editing methods, including high-throughput screening platforms, cell membrane-coated nanoparticles, and receptor–ligand interaction-mediated targeting.

### 3.1. Ligand-Mediated Targeting

Ligand-mediated targeting of nanoparticles to specific organs or cells is an attractive strategy to improve the efficiency of gene delivery. Ligands enhance the specific internalization of delivery systems through the interfacial interaction between the nanoparticles and target cell surface receptors. For example, targeting the delivery of gene therapy to central nervous system (CNS) disorders (e.g., Parkinson’s disease, Alzheimer’s disease, and Huntington’s disease) and intracranial tumors remains a challenge because the efficient delivery of gene-editing tools bypasses the blood–brain barrier (BBB) [98]. To overcome this limitation, various targeting ligands applied to engineered nanoparticles can facilitate their penetration through the BBB. With the study of the interaction between liver cells and nanomedicines, a vast number of nanoparticles based on specific ligand–receptor interactions have been designed to target different types of liver cells, such as galactose, scavenger, low-density lipoprotein (LDL), and α_V_β_3_ integrin [99]. Furthermore, precisely targeted delivery to the lesion site in gene therapy can avoid unwanted editing caused by genotoxicity and severe side effects.

#### 3.1.1. Low-Density Lipoprotein Receptor-Related Protein-1 (LRP-1)-Mediated Targeting

Low-density lipoprotein receptor-related protein-1 (LRP-1), also known as a cluster of differentiation 91 (CD91), a type I transmembrane protein belonging to the low-density lipoprotein receptor (LDL-R) family, is expressed in many cell types, such as hepatocytes and tumor cells [100]. Apolipoprotein E (ApoE) has a high affinity with several hepatic lipoprotein receptors, including LDL-R; NPs bind to ApoE and can drive LDL-R-mediated endocytosis in hepatocytes [101]. Angiopep-2 also has a high affinity to LRP-1, which is usually more highly expressed in glioblastoma (GBM) cells and brain endothelial cells (BECs). Angiopep-2-functionalized nanocapsules were shown to have 3.2-fold greater uptake in U87MG cells than nontargeting control lacking angiopep-2 nanocapsules [54].

#### 3.1.2. CD44 Receptor-Mediated Targeting

CD44, a family of non-kinase, single-span transmembrane glycoproteins, is highly expressed in many cancer cells. Hyaluronic acid (HA) is the most specific ligand for CD44 [102]. NPs functionalized with HA can selectively bind to CD44 receptors, allowing for targeted delivery of gene-editing tools to tumor cells [63]. Wang and coworkers designed a multistage-sensitive nanocomplex (MUSE), and YOYO-1-labeled MUSE performed stronger fluorescence intensity in B16F10 cells due to the HA backbone of the shell [56]. In a separate study, HA was coated on the surface of PDP by electrostatic adsorption to obtain PDA/DEX-PEI@HA (PDPH), which was verified to have enhanced tumor-targeting ability, and the internalization of nanocarriers was greatly inhibited after blocking CD44 on the surface of HeLa cells [103].

#### 3.1.3. Asialoglycoprotein Receptor-Mediated Targeting

The asialoglycoprotein receptor (ASGPR) is a C-type lectin that is highly expressed on the surface of hepatocytes. Gold nanoclusters were modified with 4-aminophenyl β-D-galactopyranoside (Gal)-modified polyethylene glycol phospholipid (Gal-PEG-DSPE), which performed much lower cellular uptake in different non-liver cell lines than Hepa 1-6 cells did. The principal reason is that galactose targets the ASGPR on the surface of hepatocytes [29]. In another study, ARPs decorated with galactose (termed Gal-ARPs) improved the ability to target the liver for the treatment of chronic hepatitis [26]. HA-PEI and HA-PEI-mannose nanoparticles with differential mannose densities (1× and 2×) were evaluated for their systemic biodistribution and hepatocyte-specific gene editing with CRISPR-Cas9. HA-PEI-mannose nanoparticles showed 55~65% uptake by hepatocytes, as well as uptake by resident macrophages, regardless of the mannose concentration [104].

#### 3.1.4. Transferrin Receptor-Mediated Targeting

Transferrin receptors (TfRs) are a class of iron-binding proteins specifically for transferrin. There are two types of transferrin receptors: TfR1 (CD71) and TfR2 (CD77), predominantly expressed in hepatocytes. TfR1 has been identified on BECs [105]. Cationic liposomes were decorated with transferrin receptor targeting peptides (THRPPMWSPVWP), and 1.5-fold higher cellular internalization was visualized in brain endothelial bEND.3 cells than in unconjugated liposomes. The targeted liposomes can significantly knock down P-glycoprotein (P-gp) expression in bEND.3 cells [106].

#### 3.1.5. Folate Receptor-Mediated Targeting

Folate receptor alpha (FR α) is a membrane-bound transport protein that is overexpressed in a range of solid tumors, making it an attractive target for cancer treatment [107]. Folic acid and its derivatives have a high affinity for folate receptors (FRs), can bind to nanoparticles to improve their transfection efficiency and selectivity in vitro and in vivo, and have potential for clinical application because of their advantages of stability and non-immunogenicity. Lin et al. mixed Cas9/sgRNA RNP complexes with oligomer #1445 to obtain a nanocarrier core and then functionalized them with FolA-PEG_24_-DBCO or PEG_24_-DBCO via click chemistry to generate FolA-PEG or PEG-modified gene-editing systems. Compared with unmodified and PEG-modified nanocarriers, FolA-PEG-modified RNP nanocarriers have greater cellular uptake ability and transfection efficiency in FRα-positive colon carcinoma CT26-EGFP/luc and cervix carcinoma HeLa-EGFP/tub cells [19]. Owing to the high expression of folate receptors in the retina, folate potentially acts as a retinal target ligand. Folate-modified CDE (FP-CDE) for the retinal delivery of TTR-CRISPR pDNA was evaluated for its usefulness in ATTR ocular amyloidosis treatment. The cellular uptake capability of the FP-CDE/fluorescein-pDNA complex changes in a folate-concentration-dependent manner, suggesting that ARPE-19 cells through FR-α-mediated endocytosis [108].

#### 3.1.6. Sialic Acid (SA)-Mediated Targeting

Phenylboronic acid (PBA) and its derivatives can recognize sialic acid (SA) to form a stable borate ester with SA, which is usually highly expressed on the surface of tumor cells. Phenylboronic acid (PBA)-functionalized disulfide bonded branched polyaminoglycoside (SS-HPT-P) has been designed to be a tumor-targeting vector of the CRISPR-Cas9 system. PBA-functionalized SS-HPT (SS-HPT-P) shows higher transfection efficiency in A549 cells than HEK293 cells due to the content of sialic acid on the surface of A549 cells is indeed significantly higher than that on the surface of HEK 293 cells [20]. Similarly, Tang et al. designed a cationic lipid (PBA-BADP), which shows 300-fold higher luciferase expression in HeLa cells than in CCC-HPF-1 cells and cancer cell-selective mRNA delivery and CRISPR-Cas9 genome editing achieved [109].

#### 3.1.7. RPE-ATRA-Mediated Targeting

The murine retinal pigmented epithelium (RPE) is a basic component of the retina and plays a crucial role in visual function. The impaired structure and function of the RPE lead to a variety of retinal diseases. Conjugated all-trans retinoic acid (ATRA), a target ligand on the surface of SMOF NPs, can enhance the RPE-specific internalization. The mechanism is that ATRA binds to the interphotoreceptor retinoid-binding protein, which selectively transports all-trans-retinol to the RPE [31].

#### 3.1.8. Integrin Receptor-Mediated Targeting

The RGD peptide (Arg-Gly-Asp) is an oligopeptide with a high affinity for the transmembrane heterodimeric αvβ3 integrin receptor, which is overexpressed on most cancer cells but downregulated on most healthy cells [110]. NPs functionalized with RGD ligands can selectively bind to integrin receptors, allowing for targeted delivery of cancer cells in tumor treatment [111]. Wang et al. co-incubated hyperbranched polyamide amine (HPAA) with c(RGDy)K to synthesize HPAA-RGD, which is better able than HAPP to enter HNE-1 cells because it specifically binds to the αvβ3 receptor on the surface of HNE-1 cells [50]. Similarly, internalized RGD (iRGD), a modified form of RGD peptide also a tumor-targeting ligand. Liposome-templated hydrogel nanoparticles (LHNPs) conjugated with iRGD in 2.6-fold increased tumor accumulation in mice [91]. In another study, RGD peptides were coupled to polycaprolactone nanofibers coated with self-assembling peptides (SAPs) to mediate the local targeted delivery of CRISPR-Cas9 to target the GDNF gene, and GDNF expression was activated, resulting in the production of biologically active GDNF to stimulate nerve regeneration. Owing to the incorporation of RGD peptides, the scaffolds could also support cell adhesion and proliferation [112].

### 3.2. Biomimetic Strategies

As emerging biomimetic nanoplatforms, cell membrane-coated nanoparticles have been developed for targeted bioimaging, drug delivery, and cancer therapy in recent years [113]. An important feature of this strategy is that membrane-coated nanoparticles indeed mimic source cells and improve the therapeutic efficacy of cargos via specific delivery to increase their accumulation in target cells and prolong their circulation in vivo.

#### 3.2.1. Coating with Cancer Cell Membrane

Different types of cell membrane coatings and core nanomaterials provide support for different biological applications. Coating with a cancer cell membrane could significantly enhance cell-specific gene-editing selectivity, which is entrusted by the inherent homotypic binding phenomenon of tumor cells. Cell somes (CSMs) derived from adenocarcinoma human alveolar basal epithelial cells (A549 cells), combined with aromatic-labeled (DOPE-647-N) and cationic lipids (DOTAP), show higher affinity toward A549 cancer cells than other cells do because of their tumor-homing effect in vitro and allow for circumventing lysosomal storage, releasing the CRISPR-Cas9 complex in the cytoplasm directly [79]. Similarly, mixing ZIF-8 with human breast adenocarcinoma cell (MCF-7)-derived membrane led to MCF-7 cell-type-specific delivery of CRISPR-Cas9 gene editing. The biological complexity of the original cell surface and its properties result in increased efficiency in cells of the same origin [114]. Compared with other cells, cell-membrane-derived vesicles (CMs) from HepG2.2.15 were coated onto the UCNPs-Cas9 surface, and the obtained UCNPs-Cas9@CM_2.2.15_ preferentially and quickly accumulated in HepG2.2.15 cells. When mouse liver cell membrane fragments form UCNPs-Cas9@CM nanoparticles, it is mainly distributed in the liver due to a good homing capability in vivo [55].

#### 3.2.2. Coating with Macrophage Membrane

In addition to the cancer cell-derived membrane, macrophage membrane-coated nanoparticles have also been explored for the delivery of the CRISPR-Cas9 system. Ping et al. extracted RAW264.7 cell membrane used for coating PD/P, and the macrophage membrane can directly guide the targeted delivery of the plasmids to the location of liver inflammatory lesions to facilitate liver-specific editing when they are administered systemically [80]. In a recent report, by taking advantage of the integrated properties of both the original cell membrane and the targeting ligands, further functionalized cell membranes showed enhanced targeting ability. Gu et al. coated mesoporous polydopamine (MPDA) with Angiopep-2-modified macrophage membranes (MMs) to form Ang-MMsaNPs, which showed significantly more efficient BBB permeation and a greater ability to target LN229 cells than MPDA-NPs [61].

To date, the cell membrane cloaking technology has provided a beneficial strategy for constructing versatile nanoplatforms for the loading and targeted delivery of several therapeutic cargos. Multiple types of cell membranes, including stem cells, red blood cells (RBCs), neutrophil cells, leukocyte cells, etc., have been separated and used as coatings in other fields [115]. Membrane vesicles and core nanoparticles can be randomly replaced and combined to form a variety of membrane-coated nanoparticles to match the needs of cell-specific targeting. Further modification of the isolated cell membranes and more precise design of the core nanocarriers will provide a new route for the transformation of biomimetic platforms into clinical applications.

### 3.3. Selective Organ-Targeting (SORT) Nanoparticles

Traditional four-component LNP systems based on cationic ionizable lipids, phospholipids, cholesterol, and PEG-lipids have been extensively studied as a gene delivery platform [116]. As the functions of the four components are further studied and clarified, a more precise design can be carried out for the structure and content of each component to meet different application requirements. Benefiting from a high-throughput platform, polymeric nano-formulation based on the combination of specific monomers is identified to enable specific cell or organ targeting without requiring the incorporation of targeting ligands and is much more effective than [117,118,119,120].

The charge of LNPs may be a key factor in determining their ability to selectively target tissues and organs. Researchers have reported a new strategy termed selective organ-targeting (SORT) nanoparticles for tissue-specific CRISPR-Cas9 delivery. They augmented conventional four-component LNPs with a fifth component (termed SORT molecule) to modulate charge to change the LNPs’ organ-targeting properties in vivo. Targeted CRISPR-Cas9 delivery to the liver and extrahepatic organs (spleen and lungs) was achieved with different supplemental SORT molecules and molar ratios [119]. Similarly, referring to the formulation protocol of SORT, LNP-assisted RNP delivery generates genome editing in both mouse liver and lungs [120]. The promise of lung-targeted therapeutic genome editing relies on the efficient delivery of gene-editing machinery bypassing mucus barrier and mucociliary clearance. There are research reports shown that replacing DOPE with the cationic lipid DOTAP can effectively improve the delivery efficiency of lipid nanoparticles (LNPs) in the lungs after systemic administration [69]. Wei and co-workers screened a series of permanently cationic lipids to optimize LNP for higher lung-specific activity. The results showed that DOTAP40 and DDAB30 have more remarkable lung targeting specificity in vivo than others, and the optimized DOTAP40 LNPs could successfully correct the CFTR mutations in cystic fibrosis models [87]. In another recent study, a library of 720 new biodegradable ionizable lipids was synthesized and screened for pulmonary mRNA delivery, among which RCB-4-8 was identified. After replacing DOPE with DOTAP, the transfection efficacy of RCB-4-8-LNPs in the lung was significantly increased, which was 100-fold higher than that of DLin-MC3-DMA (MC3), an ionizable lipid approved by the U.S. FDA for RNA delivery [69]. The existing organ-targeted LNP formulations have been summarized in Table 3, and the chemical structures of the components are shown in Figure 2. It is worth noting that current nano-delivery systems predominantly accumulate in clearance organs (e.g., liver, spleen, and kidneys) due to reticuloendothelial system (RES) sequestration and endothelial heterogeneity. To achieve precise extrahepatic delivery, future designs should synergistically integrate dynamic targeting strategies with systematic characterization of nanoparticle–bio-interface interactions. Specifically, enhanced extrahepatic targeting efficiency can be realized through tunable size/shape engineering (e.g., rod-shaped nanoparticles), stimuli-responsive targeting systems, and membrane coatings, complemented by mechanistic studies on protein corona evolution and tissue-specific endothelial trafficking pathways [121].

## 4. The Controlled Release Strategies for CRISPR-Cas9 Delivery

Controlling the self-assembly of nanoparticles, particularly in response to the cellular microenvironment to promote cargo release inside cells, is urgently needed for advancing NPs-based CRISPR-Cas9 genetic therapy. Spatiotemporal delivery vehicles have been widely developed for the controlled release of CRISPR-Cas9, similar to drugs and small nucleic acid delivery systems, and most of the structures can be precisely controlled through simple reaction steps, which can provide a versatile method for the preparation and optimization of stimuli-responsive platforms [124]. The mechanism by which they function is usually through the change or breaking of chemical bonds in the stimulating environment. An important issue to consider when designing stimuli-responsive nanoparticles is their chemosensitivity. The changeable surface of the NPs is customizable via functional groups for the controlled delivery of CRISPR-Cas9.

### 4.1. pH-Responsive

By taking advantage of pH variations among healthy tissues/tumor tissues and intracellular organelles, researchers have extensively investigated pH-sensitive delivery systems. These studies have focused primarily on utilizing the low (acidic) pH of tumor tissues and endosomes as triggers for activating delivery systems [92,105,125,126,127]. A notable feature of many pH-responsive NPs is the ability to change from an assembled state to a free state at different pH values, which enables the release of the cargo and endosome escape. The behavior can conveniently be modulated using pH-responsive groups. For example, cationic polymers, including poly (β-amino ester), PEI, etc., are important pH-responsive polymers that usually contain tertiary amine groups [48,63,65,86]. The excellent pH-buffering capacity of the imidazole bridging ligand confers pH-responsive capabilities to the ZIF, as well as an enhanced ability to escape the endocytic pathway [30,31,33].

### 4.2. Reactive Oxygen Species-Responsive

Reactive oxygen species (ROS) typically denote a category of oxygen-based chemical substances generated within the human body, including hydrogen peroxide (H_2_O_2_), singlet oxygen (^1^O_2_), superoxide, and hydroxyl radicals (HO•), which may transform from one to another via a cascade of reactions, which play important roles in signal transduction and metabolism [128]. However, the over-production of ROS in cells or tissues often leads to oxidative stress, which has implications for a series of diseases, including cancer, aging, atherosclerosis, and inflammation. The continuous production of high-level ROS by cancer cells provides various ideas for the intelligent design of ROS-responsive nanoparticle delivery systems. ROS-responsive materials usually contain reducing groups that can react with oxidizing substances, such as thioether bonds and thioketal linkers. Dithiothreitol (DTT) as a cross-link, plasmids, linear PBAE, and aPBAE (PBAE with acrylate end group) were prepared to form a post-cross-linked polyplex NPs via thiol click chemistry between the thiol of DTT and the acrylate of aPBAE, which are stable in body fluids and the thioether bond is oxidized by ROS to form sulfone groups in cells to accelerate the hydrolysis of ester bonds to remove cross-links, thereby promoting the release of loaded CRISPR-Cas9 plasmids [82].

### 4.3. GSH-Responsive

Like the pH field, GSH is an internal stimulus that can be delivered instantly to precise locations and closed systems, as it has been confirmed that the level of intracellular glutathione (GSH) is 100~1000 times higher than that of extracellular GSH. GSH can serve as a trigger for the controlled release of nanocarriers both in vitro and in vivo. Decorated with disulfide bonds is the most common strategy for designing GSH smart responsive loading systems, which can release cargo on-site via disulfide cleavage. N, N’-Bis(acryloyl)cystamine and N,N’-methylene bisacrylamide were used as different cross-linkers, and disulfide-cross-linked nanocapsules containing Cas9 and sgRNA showed 11.7-fold greater higher intracellular release than lacking disulfide bonds nanocapsules [54]. The cationic block copolymer, poly(aspartic acid-(2-aminoethyl disulfide)-(4-imidazole carboxylic acid))-poly(ethyleneglycol)(P(Asp-AED-ICA)-PEG), which bears imidazole residues and disulfide bonds, which can efficiently deliver CRISPR-Cas9 complexes in the form of DNA, mRNA, or RNP, and easily be degraded at GSH concentration to release its payload [97]. In the design of an LNP, an artificial lipopeptide, GD-LP, with a disulfide bond can be degraded in the cytosol, achieving efficient gene editing in vitro and in vivo, with negligible cytotoxicity [51]. HPAA, which consists of abundant disulfide bonds, can also be investigated as a GSH-responsive CRISPR-Cas9 delivery vector for tumor treatment [50].

### 4.4. Adenosine Triphosphate-Responsive

As the primary energy source of cellular signaling and activity, adenosine triphosphate (ATP) is the fundamental molecule that transports chemical energy within cells to support metabolism. The intracellular ATP concentration is approximately 1000-fold greater than that in the extracellular environment. In addition, the unlimited proliferation of tumor cells requires a large amount of energy to maintain growth metabolism, resulting in different ATP concentrations inside and outside the tumor cells [129]. By utilizing significant differences in ATP levels, ATP-mediated controlled release systems have been designed. A cytosolic protein delivery nanoplatform, which forms by self-assembly of imidazole-2-carboxaldehyde (2-ICA) and Zn^2+^ with the Cas9 protein, was disintegrated to release the protein in the presence of a physiological concentration of ATP, as a result of the competitive coordination between ATP and the Zn^2+^ of ZIF-90, while the efficient delivery of Cas9/sgRNA and on-target cleavage of the GFP gene led to the knock-out of the green fluorescent protein (GFP) expression in HeLa-GFP cells with an efficiency of up to 35%. There is no doubt that, compared with Cas9 alone, ZIF-90/Cas9 nanoparticles improve the transportation efficiency of the Cas9 protein [34].

### 4.5. Near-Infrared Light-Responsive

In the past few decades, in-depth research on the principle of the near-infrared (NIR) stimulus response at the molecular level has provided a reference and guidance for designing NIR-responsive nanocarriers for controlled release, and numerous NIR light-responsive nanoplatforms have flourished and are promising candidates for practical applications involving on-demand gene/protein/drug delivery.

Lanthanide-doped upconversion nanoparticles (UCNPs) can convert near-infrared radiation with lower energy into visible radiation with higher energy through a nonlinear optical process. NIR, responsive nanocarriers of CRISPR-Cas9 based on UCNPs, can convert NIR light into local ultraviolet light for the cleavage of photosensitive molecules, resulting in the controlled release of CRISPR-Cas9. CRISPR-Cas9 was covalently anchored onto UCNPs via photocleavable 4- (hydroxymethyl)-3-nitrobenzoic acid (ONA) molecules and then coated with PEI to obtain UCNPs-Cas9@PEI, which can release Cas9 protein into the nucleus upon NIR irradiation, but it only appears in the cytoplasm and scarcely appears in nuclei without the irradiation of NIR light, proving that nanocarriers have the potential for NIR light-activated genome editing both in cells and in deep tissues [84]. As the same principle, photostable PC biotin-NHS ester (PCB) was used to assemble the UCNP-Cas9 complex, which was then coated with CMs onto the surface to get NIR-responsive biomimetic nanoparticles (UCNPs-Cas9@CM), which can release CRISPR-Cas9 under NIR light irradiation to disrupt HBV cDNA to inhibit viral replication [55].

The semiconductor copper sulfide (CuS) is one of the most widely studied inorganic photothermal therapy (PTT) reagents and has served as a vehicle for CRISPR complexes. In accordance with the principle of base complementarity, the RNP complex was applied to assemble with CuS-DNA to form CuS-RNP@PEI, which can release RNP into the cytoplasm via a double-chain break induced by optothermal radiation provided by NIR-triggered CuS; moreover, tumor ablation and ICD was induced upon NIR-triggered photothermal ablation (∼47 °C) via CuS NPs [86].

### 4.6. Magnetic-Responsive

Magnetic nanoparticles, a rapidly developing new type of functional material, have unique magnetic properties that can respond to different external magnetic fields, generating different effects, such as force and heat.

Fe_3_O_4_ NPs, well-known traditional magnetic nanomaterials, are usually used to endow nanocarriers with magnetic responsive properties. Encapsulating Fe_3_O_4_ NPs in mesoporous polydopamine (mPDA) as the core can provide magnetic targeting ability and promote the accumulation of nanocarriers in the tumor site under the stimulation of an external magnetic field [36]. Zhu et al. complexed magnetic iron oxide nanoparticles (MNPs) with recombinant baculoviral vectors (MNP-BVs) as a CRISPR-Cas9 delivery vector, and gene editing was activated locally in vivo via a magnetic field. An external magnetic field can serve as an ‘on’ switch for tissue-specific genome editing for spatial control [37]. In a recent study, researchers synthesized ZnFe_2_O_4_ cores via the thermal decomposition of a mixture of metal precursors and constructed a novel magnetic nanoparticle-assisted genome-editing (MAGE) platform via layer-by-layer assembly. Compared with Fe_2_O_3_ or Fe_3_O_4_ cores, ZnFe_2_O_4_ cores have a higher saturation magnetization. The magnetofection can first improve the delivery of multiple plasmids, and plasmids-containing cells can then be purified by magnetic-activated cell sorting (MACS), effectively delivering CRISPR plasmids to iPSC NPCs and monitoring cell uptake [38].

### 4.7. Dual Stimuli-Responsive and Multistage-Responsive

The pathological area of the disease is markedly different from the normal tissue environment and is compounded by the concurrent presence of various stimuli, making it challenging for stimuli-responsive delivery systems to deliver high concentrations of cargo to the target site precisely and rapidly. Consequently, designing and developing stimuli-responsive nanoplatforms that can respond to two or more signal combinations is expected to further improve the performance of controlled release.

pH/GSH-responsive nanoparticles are among the most widely studied classes. The cationic block copolymer, poly (aspartic acid-(2-aminoethyl disulfide)—(4-imidazole-carboxylic acid))-poly (ethylene glycol) (P(Asp-AED-ICA)-PEG) was developed as a dual stimuli-responsive nanoplatform for DNA, mRNA, and RNP delivery. The imidazole residues confer pH-responsive capabilities, and disulfide bonds can be selectively cleaved in the cytosol because of the relatively high concentration of intracellular GSH, resulting in the release of the payload. The results confirmed that P(Asp-AED-ICA)-PEG shows higher transfection efficiency and lower cell toxicity than Lipofectamine 2000 in most cell lines [97]. Li and coworkers proposed a pH/GSH-dual responsive nanoreactor to controllably co-deliver Ca^2+^, CO gas and RNP. Calcium carbonate (CaCO_3_) coated silica spheres can be biodegradable by pH-induced hydrolysis, releasing glucose oxidase (GOx) to catalyze glucose for H_2_O_2_ production, which further reacts with manganese carbonyl (MnCO) and achieves the precise release of CO gas to cause metabolic exhaustion in cancer cells. Simultaneously, in situ Ca^2+^ overload from CaCO_3_ degradation interferes with mitochondrial Ca ^2+^ homeostasis. As a result, Ca^2+^-driven ROS generation and subsequent mitochondrial apoptosis signaling pathways are activated. At the same time, RNP is released for efficient gene editing because of disulfide bond cleavage in DSP induced by GSH. MG-RNP@CaCO_3_ could effectively inhibit tumor growth and protect normal surrounding tissue from oxidative stress in A549 tumor-bearing mice [56].

In another case, phenylboronic acid (PBA) can selectively bind to the cis-diol moieties of common sugars, but the binding affinity depends on the pH; it is strong at physiological and basic pH but weak at acidic pH. ATP has a cis-diol moiety in its ribose ring, which interferes with the PBA–sugar interaction, resulting in the formation of PBA-ATP in acidic environments. Owing to the interaction between 4-carboxy-3-fluoro PBA (FPBA) and ATP, with pH and ATP as dual stimuli-responsive, PBA-functionalized chitosan-poly ethylenimine (CS-PEI) polymers were investigated for their ability to edit the liver, resulting in the downregulation of target proteins after oral administration [130].

Multistage-responsive nanoplatforms have been designed to overcome sequential biological barriers and efficiently deliver CRISPR-Cas9 to target sites. Yang et al. reported a programmable unlocking nano-matryoshka-CRISPR system (PUN) in response to tumor microenvironment (TME), which own hierarchical multistage responsiveness for precise and efficient control of CRISPR-Cas9 release and editing. To construct such a TME multistage-responsive system, PEI 1.8K was crosslinked with an ROS-cleavable thioketal linker, and then, the dual-enzyme-responsive copolymer hyaluronic acid (HA)-RGD peptide (Arg-Gly-Asp)-MMP substrate (GPLGVRG)-polyethylene glycol (PEG) (HRMP) was prepared through a two-step reaction. The first unlocking process at the MMP-rich tumor microenvironment enhances tumor recognition, deep penetration, and cellular internalization depending on exposed RGD and HA. The second unlocking is completed in lyso/endosomes, in which HAase triggers a charge reversion by degrading HA to facilitate rapid lysosomal escape. High endogenous ROS turns on the last unlocking for effective release of payload [83]. Lu et al. developed triple functional nanoparticles (Fe_3_O_4_@mPDA-mPEG-Ni, FPP) by encapsulating Fe_3_O4 nanoparticles with a layer of mesoporous polydopamine (mPDA) for delivery of Cas9 RNP-targeting PD-L1 gene. The magnetic targeting effect of Fe_3_O_4_ NPs can enhance the accumulation of Cas9 RNP-loaded nanoparticles at the tumor sites, and the photothermal effect of PDA can generate mild PPT to induce ICD, resulting in the generation of tumor-associated antigens that help enhance anti-tumor immune responses [36].

## 5. The Application of Delivery of the CRISPR-Cas9 by NPs

A variety of nanomedicines have already been approved by the Food and Drug Administration (FDA), which indicates that gene-editing therapy is destined to enter the therapeutic area. CRISPR-Cas9 genome editing with nanoparticles as vectors could potentially treat the root causes of many genetic diseases. Dramatically, the results of the first in vivo clinical trial of CRISPR gene-editing therapy were revealed, greatly expanding the application scope of CRISPR gene-editing therapy. Many potential therapeutic pathways are still under investigation.

### 5.1. Gene Therapy

#### 5.1.1. For Genetic Disease Therapy

The emergent technology of gene editing, which is based on CRISPR-Cas9, has greatly powered genetic disease therapy-associated studies, especially for monogenic diseases. CRISPR-Cas9, which is based on chemically synthesized nanocarriers for gene therapy of monogenic genetic diseases, has been investigated in animal models in recent years.

Duchenne muscular dystrophy (DMD) is an inherited X-linked disease caused by a loss-of-function mutation in the gene encoding dystrophin; the dystrophin protein plays a key role in skeletal muscle fiber integrity [127]. CRISPR-Cas9-based gene therapy is a potential candidate, and it has been studied in mouse models [131]. Guanidium-rich lipopeptide GD-LP-mediated delivery of RNP/ssODN targeting the DMD gene in mdx mice showed that the RNP/ssODN-LNP treatment sufficiently restored dystrophin expression, reduced skeletal muscle fibrosis, and improved muscle strength [51]. In another remarkable study, an LNP-delivery system was proposed to preferentially target skeletal muscle. LNPs can achieve accumulative dystrophin protein recovery by repeated administration over time and with low immunogenicity. Furthermore, administration of the LNP via the limb perfusion method was developed to enable the targeting of broader muscle groups [86]. GNPs were also devised to co-deliver RNP and donor DNA, which effectively induce HDR and correct DNA mutations in muscle tissue that cause Duchenne muscular dystrophy in mdx mice via local injection, minimizing off-target DNA damage [28].

β-hemoglobinopathies are severe human autosomal monogenic inherited diseases, mainly β-thalassemia and sickle cell disease (SCD), and allogeneic hematopoietic stem cell (HSC) transplantation is currently the only definitive treatment option [132]. Targeting the γ-globin gene locus can efficiently change the expression of fetal hemoglobin (HbF). Biodegradable and FDA-approved poly(lactic-co-glycolic acid) (PLGA) linked with PEG as delivery vehicles for the CRISPR-Cas9 complex can be taken up by HUDEP-2 cells, primary erythroblasts, and CD34 cells, achieving robust gene editing to increase levels of HbF without biological toxicity [21]. Bone-marrow-homing lipid nanoparticles (BM-homing LNPs) deliver a CRISPR-Cas9 gene-editing system to target the HBG1/HBG2 gene that could induce HbF in a mouse model and has the potential to reduce the morbidity and mortality of β-hemoglobinopathies [133].

Hemophilia A and B are the most common severe hereditary hemorrhagic disorders and are caused by factor VIII and factor IX protein deficiency. CRISPR-Cas9-mediated genome editing for hereditary hematological disorder therapy has been widely explored in recent years [134]. Antithrombin (AT), an endogenous negative regulator of thrombin generation, is a potent genome-editing target for sustainable treatment. A novel approach, LNP-CRISPR-mAT treatment, leads to the inhibition of AT, resulting in an improvement in thrombin generation. The bleeding-related phenotypes of hemophilia A and B mice both recovered and did not cause liver damage or detectable off-target effects [15]. Notably, LNP-mediated delivery of CRISPR-Cas9 gene editing for sickle cell disease and β-thalassemia has shown promising results in clinical trials. Although adverse events were reported in both patients after the CTX001 infusion, this landmark study demonstrates the clinical translation potential of CRISPR-Cas9-mediated gene therapy [135].

Transthyretin amyloidosis, also known as ATTR amyloidosis, is caused by the progressive accumulation of misfolded transthyretin (TTR) protein in tissues, predominantly the nerves and heart. Inhibiting the hepatic synthesis of TTR is a promising therapeutic option for ATTR amyloidosis [136]. LNP-INT01 has been developed as a CRISPR-Cas9 component delivery vector, achieving robust and persistent in vivo genome editing with a single administration. In particular, there was no significant difference in editing efficiency between rats and mice; the editing efficiency reached approximately 70%, while serum levels of TTR were reduced by more than 90% in the livers of rats, showing the potential for preclinical applications [16]. Encouraged by the results of preclinical studies in the mouse and cynomolgus monkeys, the CRISPR-Cas9 approach used for LNP-NTLA-2001 has been approved for an ongoing phase 1 clinical study [89]. The serum TTR protein level significantly and persistently decreased after a single admission of NTLA-2001; however, while NTLA-2001 is generally well tolerated, further research on its long-term safety and efficacy is still necessary [137].

These promising strategies could also be adjusted for application in studies of other similar genetic disorders, even though genome editing within the germ line is currently not feasible in humans. However, genome editing can be envisioned in postnatal cells in vivo in principle after overcoming the key technical challenges. Further development and evaluation are required to ensure the effectiveness and safety of these methods. Furthermore, the safety of the CRISPR-Cas9 system, especially for long-term treatment, needs to be evaluated in preclinical studies of large animal disease models.

#### 5.1.2. For Cancer Therapy

Another important application of CRISPR-Cas9 lies in the effective knock-out of the oncogenic gene for cancer therapy.

The serine protease proprotein convertase subtilisin/kexin type 9 (PCSK9), a protein closely related to cholesterol regulation, plays an important role in various diseases, including cardiovascular diseases [126]. There are currently multiple treatment strategies for treating hypercholesterolemia and other related diseases with PCSK9. Jiang et al. designed triple-targeting gold nanoclusters, achieving efficient gene editing of PCSK9 in the liver in approximately 60% of cases and resulting in the downregulation of serum low-density lipoprotein cholesterol [29].

Polo-like kinase 1 (PLK1) plays a pivotal role in cell division and is commonly overexpressed in various types of tumors, making it a model therapeutic gene that has been widely used to evaluate the efficiency of delivery vectors to inhibit tumor growth and improve tumor-bearing mouse survival [138]. Glioblastoma is the most common and aggressive form of solid cancer in the central nervous system. The targeted knock-out of the PLK1 gene and protein reduction, increase the median survival time of mice bearing orthotopic glioblastoma by approximately 44 days. LHNPs, as versatile CRISPR-Cas9 delivery tools, resulted in a 60.4% decrease in PKL1 expression, which effectively prolonged the median survival time of mice bearing intracranial U87 gliomas from 29 days to 40 days [91]. UCNPs-Cas9@PEI targeting the PLK-1 gene successfully inhibited cancer cell proliferation and tumor growth without noticeable abnormalities or appreciable organ damage to major organs from different groups of mice [84].

Latent membrane protein 1 (LMP1), encoded by the typical oncogene Lmp1, is an important factor in virus–host interactions and a potential therapeutic target. Downregulation of LMP1 can promote the proliferation and metastasis of nasopharyngeal carcinoma cells by regulating the downstream signaling pathways of p38, NF-kB, and PI3K [139]. The P-aP-DTT-LMP-g4 polyplex NP system targets the Lmp1 oncogene to downregulate the expression of LMP1 protein, which displays antitumor activity both in vitro and in vivo, indicating the potential for gene therapy in EBV-related nasopharyngeal carcinoma [82].

The Fas gene is an apoptosis-regulated gene related to the progression of chronic hepatitis, and cationic polymer-coated AuNRs have been reported to target the Fas gene to prevent the progression of chronic hepatitis. After tail-vein injection of the Gal-ARP8/CMV-Cas9-Fas complex, the levels of indicators of hepatic fibrosis, hydroxyproline, and procollagen III are downregulated. Furthermore, it effectively reduces the degree of hepatic fibrosis or necrosis [26].

### 5.2. Multimodal Synergistic Cancer Therapy

Although existing results have shown that CRISPR-Cas9-mediated gene therapy can inhibit tumor growth and reduce the metastatic burden, single gene-targeted therapy cannot completely eliminate tumors due to the complexity, diversity, and heterogeneity of the tumor microenvironment. The remaining un-killed cells may lead to tumor relapse, highlighting the limitations of CRISPR-Cas9 gene editing in cancer gene therapy [43,140]. Current research has gradually shifted from monotherapy to combination therapy, and the collaborative and augmentative interactions among diverse types of monotherapies have led to the emergence of multimodal synergistic therapy. In turn, extraordinary super-additive effects, denoted as “1 + 1 > 2”, which surpass the potency of any single therapy or their anticipated combined efficacy in theory, are engendered [141]. For example, CRISPR-Cas9-mediated knock-out of heat shock protein 90 (Hsp90) effectively reduces the thermal resistance of cancer cells, resulting in a mild NIR-light-triggered PTT [85,140].

## 6. Challenges and Future Directions

In recent years, genome editing has entered a boom period because of its broad and effective application prospects in scientific research and disease treatment. Among all genome-editing tools, the CRISPR-Cas9 system is the most flexible, convenient, and widely used. The combination of CRISPR technology and chemically synthesized, non-viral vector-based drug delivery systems has been proven to be a powerful way to efficiently deliver CRISPR components to target cells or tissues, enabling the specificity and accuracy of genome editing required for safer gene therapies. The delivery of CRISPR-Cas9 via nanocarriers has great potential for clinical application. There are certain similarities in the research of different nanoparticles, especially for improvement and modification strategies, which are worthy of in-depth study and mutual reference. However, there are still many obstacles to overcome in order to expedite clinical translation.

### 6.1. Clinical Translation Challenges

#### 6.1.1. Manufacturing Barriers

First of all, large-scale production of nanoparticles also faces some commercial barriers, such as batch-to-batch variability in scalable synthesis, NPs aggregation during sterile filtration, and mRNA degradation under low-temperature preservation [142]. Furthermore, batch-to-batch variation is widespread throughout some production processes of nanomedicines [143,144]. DepoCyte, a liposomal suspension for injection containing the anti-cancer agent cytarabine, has had its application for variation to marketing authorisation withdrawn. This decision was mainly due to concerns over sterility and reproducibility issues [145], highlighting the importance of ensuring both product sterility and reliable data reproducibility in the pharmaceutical development and approval process.

#### 6.1.2. Regulatory Hurdles

Furthermore, great attention should be paid to the regulatory framework of the FDA and EMA regarding gene-editing therapies. The World Health Organization (WHO), FDA, and European Medicines Agency (EMA) have established evolving guidelines for CRISPR-based therapies, emphasizing three core requirements: recision validation, long-term safety, and manufacturing control [146,147,148]. Recently, a case study has shown that the ambiguities surrounding the definitions of the active ingredient and excipients for nanomedicines can exert a notable influence on their approval process [149].

#### 6.1.3. Ongoing Clinical Trials and Technical Hurdles

After the world’s first CRISPR therapy, which aims to cure sickle cell disease and transfusion-dependent β-thalassemia is approved, the relatively dispersed nature of the patient population suffering from this disease, the high costs involved, and the concerns of patients and their families towards novel therapies have jointly contributed to the sluggish enrollment progress in clinical trials. This situation has severely hampered the pace of clinical translation [150]. In addition, current clinical trials have provided clinical proof of concept as a gene therapy strategy. However, the subjects still need to undergo long-term safety monitoring, and many more safety issues remain unknown [89]. In addition, the occurrence of challenges in other comparable studies has the potential to impact the project’s progress. This is exemplified by the case of bluebird bio. The company has temporarily suspended its Phase 1/2 (HGB—206) and Phase 3 (HGB—210) studies of LentiGlobin gene therapy for sickle cell disease (SCD) (bb1111) as a result of a reported suspected unexpected serious adverse reaction (SUSAR) involving acute myeloid leukemia (AML) [151]. These challenges underscore the need for standardized manufacturing protocols and adaptive regulatory frameworks to accelerate clinical deployment.

### 6.2. Future Directions

#### 6.2.1. Artificial Intelligence (AI)-Driven Nanoparticle Design

The integration of machine learning (ML) and generative AI is revolutionizing nanocarrier optimization. The rapid development of computational science continues to bring innovation and opportunities to the field of drug discovery. Through interdisciplinary integration and innovation, combining techniques such as machine learning, molecular docking, multi-omics analysis, and clinical trial prediction etc. [152,153,154]. Through the LNPs designed with the assistance of AI, researchers have successfully optimized the delivery system for pulmonary gene therapy [155]. One successful methodology may be further expanded to allow the rational design of nanoparticles that selectively target specific cells or tissues for the delivery of different types of drugs and additional cells or tissues in the future, contributing to accelerating clinical development. It is worth noting that the EMA issued a document advocating that when applying AI throughout the lifecycle of medicinal products, compliance with current legal obligations is essential. Ethical considerations must be factored in, and fundamental rights should be duly respected [156].

#### 6.2.2. LNP: The Trailblazer in CRISPR Gene Therapy

LNP has successfully bridged the gap from basic research to clinical translational applications, greatly propelling the development of the fields of gene therapy and vaccines after 60 years of research. Currently, a variety of vaccines and therapies utilizing LNP delivery technology have been approved by the FDA or are in the stage of clinical development [157]. The recent approvals set a precedent for CRISPR therapies; the future prospects of gene therapies enabled by LNP-based delivery systems are evidently transformative [89,158]. This is mainly attributed to their high efficiency in delivering genetic materials, low immunogenicity, and the potential to target specific cell types precisely. The LNP is continuously evolving and being optimized. In the future, it is certain to propel gene therapy technology towards broader clinical applications, bringing new hopes for the treatment of a variety of refractory diseases.

## Figures and Tables

**Figure 1 nanomaterials-15-00540-f001:**
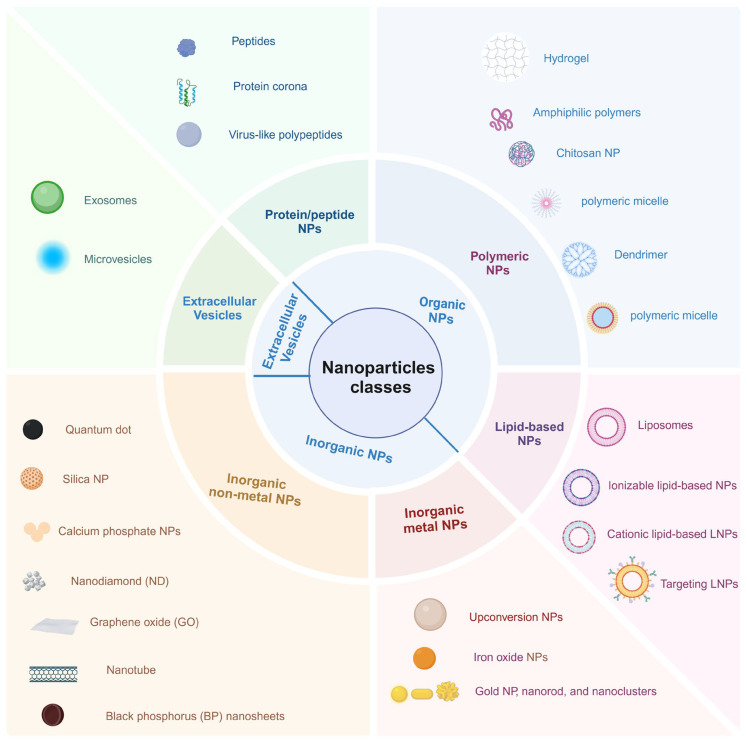
The nanoparticles used for CRISPR-Cas9 delivery.

**Figure 2 nanomaterials-15-00540-f002:**
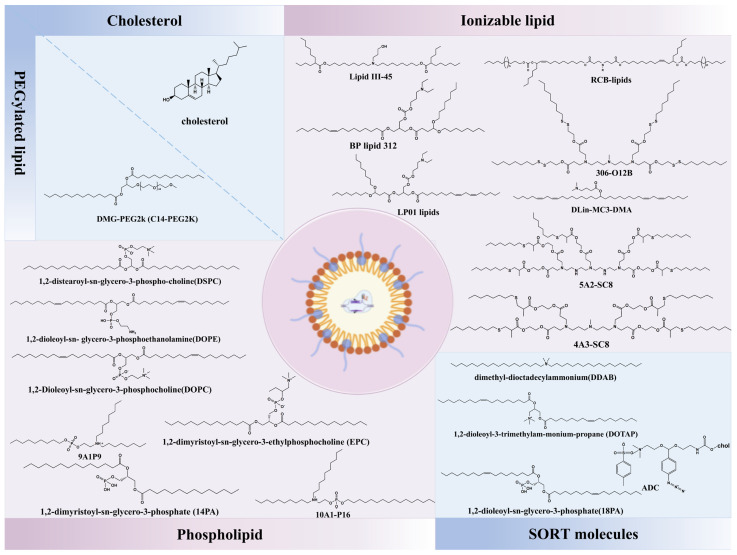
The chemical structures of lipids and lipid derivatives used for targeting CRISPR-Cas9 delivery.

**Table 1 nanomaterials-15-00540-t001:** NPs for CRISPR-Cas9 gene editing in vitro.

Materials	Hydrodynamic Diameter	Cas9 Form and Encapsulation Efficiency (or Loading Content)	Target Cell; Gene; Efficiency and Analysis of Gene Editing In Vitro	Refs.
Gold nanorods (AuNRs)	150 nm	plasmids (1.35 μg/μg)	AAVS1 in 293T cells (15.37% knock-out) and PLK1(15.50% knock-out) in A549 cells by analyzing gel bands of the digestion products of T7E1, Fas gene (10.5% knock-out) in Hepa1-6 cells by Sanger sequencing	[26]
Gold nanoparticles (GNPs)	546 nm	RNP and donor DNA (61.5%)	CXCR4 in hES cells, hiPS cells, BMDCs, and the dystrophin gene (HDR efficiency 3~4%) in myoblasts by PCR	[28]
Au nanoparticle-loaded core–shell tecto dendrimers (Au CSTDs)	108.0~131.2 nm	plasmids	PD-L1 (59.8% knock-out) in B16F10 cells by T7 endonuclease I (T7EI) assay	[24]
Protamine-capped gold nanoclusters	3.5 nm	plasmids	EGFP gene (30% knock-out) in U2OS-EGFP cells by flow cytometry, human papillomavirus (HPV) type 18 (HPV18)-E7 oncogene (29% knock-out) in HeLa by T7 endonuclease I (T7EI) assay and Western blotting	[22]
Cancer cell membrane-derived nanocarrier (mCas9-sGNRs)	length ≈ 60 nm, width ≈ 15 nm	RNP (60%)	Survivin gene (33% knock-out) in MDA-MB-231 cells by T7 endonuclease I (T7EI) assay	[43]
Cationic gold nanorod	(length 60.14 ± 3.56 nm, width 8.02 ± 0.59 nm)	plasmids (ANP/plasmids = 0.15)	PD-L1 (39.7% indel) in B16F10 cells by T7 endonuclease I (T7EI) assay	[44]
Silica–metal–organic framework hybrid nanoparticles (SMOF NPs) consisting of both silica and zeolitic imidazole framework (ZIF)	110 nm	RNP (>9 wt% and loading efficiency > 90%)	HEK 293-GFP cells (~60% knock-in) by flow cytometry	[31]
Zeolitic imidazolate framework-8 (ZIF-8)	100 nm	RNP (1.2%, *w*/*v* and loading efficiency of 17%)	EGFP gene (37% knock-down) in Chinese hamster ovary (CHO) cells by qRT-PCR	[33]
Core–shell hollow mesoporous organosilica nanoparticles	156.6 ± 1.8 nm	plasmids (76.65%)	EGFR gene (66.3% knock-out) in HepG2 cells by T7 endonuclease I (T7EI) assay	[45]
ZIF8-NaHCO_3_ @Cas9 (ZNC)	40~70 nm	plasmids	RANKL (71.12% knock-out) in MC3T3-E1 cells by flow cytometry analysis	[46]
PB@RNP-EuMOFs	~150 nm	RNP (60%)	GFP (47% knock-out) in HeLa-GFP cells by Sanger sequencing assay	[47]
Polyethylenimine (PEI) and polyethylene glycol (PEG) conjugated carbon quantum dots (CQDs-PP)	2.4 nm	plasmids	EFHD1 gene (34.2% knock-out) in HeLa cells by PCR and sequencing	[48]
PBA-rich cationic polymer	300 nm	RNP	EGFP (40% knock-out) in HEK293-EGFP reporter cells by flow cytometry	[49]
Hyperbranched polyamide amine (HPAA)	500 nm	plasmids	SGK3 gene (13% knock-out) in HNE-1 cells by flow cytometry	[50]
Lipopeptide (GD-LP)	231.3 nm	RNP (20 wt%)	EGFP gene (72.6% knock-down) in GFP-HEK 293 cells by flow cytometry	[51]
Carboxylated branched poly (b-amino ester)	~200 nm	RNP (30 *w*/*w*)	GFP gene (77% knock-out) in HEK293 cells and GFP gene (47% knock-out) in GL261 murine glioma cells	[52]
Angiopep-2 decorated, guanidinium and fluorine functionalized polymeric nanoparticle	~143 nm	RNP (50 *w*/*w*)	PLK1 gene (32% knock-out) in U87MG cells by restriction enzyme (BstAP I) digestion assay	[53]
Angiopep-2-functionalized, disulfide-cross-linked nanocapsules	~30 nm	RNP (almost 100%)	PLK1 gene (38.1% knock-down) in U87MG cells T7E1 assay	[54]
Phenylboronic acid (PBA)-functionalized, disulfide-bonded branched polyaminoglycoside (SS-HPT-P)	200 nm	plasmids (SS-HPT-P2/pDNA (*w*/*w* = 30))	Survivin gene (20% knock-out) in A549 cells by flow cytometry	[20]
CuS-RNP@PEI	28 nm	RNP (CuS/RNP (*w*/*w* = 8))	GFP (40.7% knock-out) in GFP- MDA-MB-231 cells by flow cytometry; PTPN2 gene (36.5% knock-out) in HEK 293 cells by T7 endonuclease I (T7E1) assay	[20]
UCNPs-Cas9@CM	~140 nm	RNP (~33.3%)	HBV DNA (33.75% knock-out) in HepG2.2.15 and HepAD38 cells by sequence analysis	[55]
MG-RNP@CaCO_3_	< 100 nm	RNP	Nrf2 gene (36.03% knock-out) in EGFP-A549 cells by T7 Endonuclease I (T7E1) assay	[56]
P/M@CasMTH1	~100 nm	RNP (MTK/sgRNA = 1:6)	EGFP gene (42.6% knock-out) in EGFP-A549 cells by CLMS, MTH1 gene (39.9% knock-out) in A549 cells by T7 endonuclease I (T7EI) assay	[57]
PLGA	~350 nm	49–75% for Cas9 and 69–89% for sgRNA	EGFP gene (70% knock-down) in EGFP-HUDEP-2 cells by RT-qPCR	[21]
PLGA	210~350 nm	plasmids (1.6 wt%)	Cas9 protein expression in wild-type mouse bone marrow-derived macrophages (BMDMs) by Western Blots and gene-editing efficiency was comparable to Lipofectamine	[58]
Methoxy-poly(ethyleneglycol)-b-poly(2-(azepan-1-yl) ethyl Methacrylate) (mPEG-PC7A)	~30 nm	RNP (17 wt%)	HDR efficiency of HDR-NP to 7.0% in BFP-expressing human embryonic stem cells (hESCs) by flow cytometry	[59]
Multistage-sensitive nanocomplex (MUSE)	138 ± 3 nm	plasmids	CD47 (35% indel) and PD-L1 (47% indel) in B16-F10 cells by T7 endonuclease I (T7EI) assay	[60]
pCas9-loaded nanocore (PRTM/pCas9/Ca; NP)	187 ± 3 nm	plasmids (90%)	HIF-1α (~75% knock-out) in H1299 cells by qPCR	[61]
Cas9En-ArgNP nano-assemblies	475 ± 60 nm	RNP (ArgNP: Cas9En = 2:1)	AAVS1 gene (29% knock-out) in HeLa cells byT7 endonuclease I (T7EI) assay	[25]
Peptide/lipid-associated nucleic acids (PLANAs)	~100 nm	RNP (89%)	HPRT protein (35% indel) in HEK293 cells by T7 endonuclease I (T7EI) assay	[62]
Multifunctional nanosystem (HPR@CCP)	51.8 ± 10.3	plasmids (99.8 ± 2.36%)	EGFP gene (74.12% expression) in mouse skin melanoma cells B16F10 cell line by flow cytometry	[63]
F-PC/pHCP	∼50 nm	plasmids (N/P = 2)	PD-L1 gene (58.4% knock-out) in B16F10 cells by flow cytometry	[64]
Silica nanoparticle (SNP)	52 ± 4 nm	RNP (>90%)	GFP gene (~72% knock-out) in GFP- HEK293 cells by T7 endonuclease I (T7EI) assay	[32]
Silica nanocapsules (SNCs)	∼50 nm	RNP (>90%)	GFP gene (~70% knock-out) in GFP- HEK293 cells by T7 endonuclease I (T7EI) assay	[35]
Magnetic core–shell nanoparticle (MCNP)	98.84 ± 3.96 nm	plasmids (>80%)	MeCP2 gene (42.95% repair) in induced pluripotent stem cell-derived neural progenitor cells (iPSC-NPCs) from a Rett syndrome patient by confocal laser scanning microscopy (CLSM)	[38]
Fe_3_O_4_ @mPDA-mPEG-Ni	260 nm	RNP (92.8%)	EGFP gene (45–50% knock-out) in EGFP-293 T cells by flow cytometry and PD-L1 gene (42.1% knock-out) in B16F10 cells by confocal laser scanning microscopy (CLSM)	[36]
Lipid-modified oligoamino amides and folic acid (FolA)-PEG	~50 nm	RNP (89.5%)	PD-L1 (60.7% knock-out) and PVR (58.7% knock-out) in CT26 cells by Sanger sequencing	[19]
Rolling circle amplification (RCA)-based multifunctional DNA/UCNP complex	~45 nm	plasmids (DNA layer ≈4.80 nm thick)	Nuclear factor E2-related factor 2 (Nrf2) gene (18.7% knock-out) in MCF-7 cells by agarose gel electrophoresis	[65]
Binding-mediated protein corona (BMPC)	24.2 nm	RNP corona (25:1)	EGFP (12.8% indel) and EMX1 gene (10.5% indel) in EGFP-HEK293 cells by T7 endonuclease I (T7EI) assay	[66]
4A3-SC8 dLNPs	100 nm	mRNA (>92%)	GFP (≈18% HDR efficiency) in HEK293 B/GFP cells by flow cytometry	[67]
LNP-INT01 (LP01+PEG-DMG)	<100 nm	mRNA (>95%)	TTR gene (>97%) in mouse primary hepatocytes by next-generation sequencing (NGS) analysis	[16]
LNP (P127 M@pCD98)	267.2 nm	plasmids (100%)	CD98 (61.3% knock-down) in CT-26 cells by RT-qPCR	[68]
LNP (RCB-4-8)	85.7 ± 1.6 nm	mRNA (87.1 ± 2.3%)	GFP (~95% knock-out) in HEK 293 cells by flow cytometry	[69]
LNP (iLP181)	98.43 ± 6.00 nm	plasmids	PKL-1(>30% knock-out) in HepG2-Luc cells by RT-qPCR	[70]
LNP(BAMEA-O16B)	233.6 ± 2.3 nm	mRNA	GFP (>90% knock-out) in HEK-GFP cells by confocal laser scanning microscopy (CLSM)	[18]
LNP	75.3 nm	mRNA (>90%)	SERPINC1 gene (~60 knock-out) in mouse C2C12 cell by targeted deep sequencing	[15]
LNP	80 nm	mRNA (>90%)	EGFP gene (94% knock-out) in GFP-HEK293 cells by next-generation sequencing (NGS), PKL-1(98% knock-out) in GFP-HEK293 cells by next-generation sequencing (NGS)	[71]
LNP	90 ± 4 nm	plasmids (N/P = 6)	FOXC1 gene (~80% knock-out) in MDA-MB-468 by confocal laser scanning microscopy (CLSM)	[72]
Peptide-conjugated lipids	159.80 ± 3.87 nm	plasmids (95.47 ± 4.38%)	HuR gene (48.94 ± 0.68% knock-out) in SAS cells by confocal laser scanning microscopy (CLSM)	[73]
LNP	112.5~144.0 nm	mRNA (74.6~82.9%)	CRE (53.2%~61.8% td-tomato positive cells) in NIH 3T3 CRE reporter cells by flow cytometry	[74]
Nano-cleaver (HepCCCleaver)	215.3 ± 1.1 nm	plasmids (wt/wt = 50)	HBV DNA (89.1% knock-out) in HepAD38 cells by T7 endonuclease I (T7E1) assay and DNA sequencing	[75]
LNP	79.1 nm	mRNA (96%)	DMD gene (43.6% knock-out) in DMD patient myoblasts by T7 endonuclease I (T7E1) assay	[76]
Lipid/AuNPs complex	101.2 ± 5.6 nm	plasmids (97%)	PLK-1 gene (65% knock-out) in A375 cells by Western blot assay	[77]
PBA- BADP/Cas9 mRNA NPs	111 ± 2 nm	mRNA	EGFP gene (~25% knock-out) in EGFP- HEK293 cells and (~50% knock-out) in EGFP-HeLa cells by T7 endonuclease I (T7E1) assay	[78]
PEGylated nanocapsules (NCs)	36 ± 3 nm	RNP (~40 wt%)	GFP gene (~70% knock-out) in GFP- HEK293 cells by flow cytometry	[30]
Fusogenic cancer cell-derived nanocarriers	~200 nm	RNP	EGFP (knock-out 35%) in 293-T-HEK-dEGFP reporter cells by flow cytometry	[79]
US-propelled Cas9/sgRNA@ AuNWs	400 nm	RNP	GFP (80% knock-out) in GFP-B16F10 cells by confocal laser scanning microscopy (CLSM)	[23]

**Table 2 nanomaterials-15-00540-t002:** NPs for CRISPR-Cas9 gene therapy in vivo.

Materials	Model	Target Gene	Target Disease	The Modes of Administration	CRISPR Cas9 Dose	Duration	Outcome	Refs.
Lipid-modified oligo amino amides and folic acid (FolA)-PEG	CT26 in Balb/c tumor model	PD-L1 and PVR gene	CT26 tumor	intra-tumoral injection	plasmids (12.5 μg)	every 2 days for 3 days	dual PD-L1/PVR immune checkpoint disruption, and suppressed tumor growth in vivo	[19]
Phenylboronic acid (PBA)-functionalized, disulfide-bonded branched polyaminoglycoside (SS-HPT-P)	tumor-bearing mouse model	survivin gene	tumor	tail-vein injection	plasmids (25 μg)	every 2 days for 12 days	inhibited tumor proliferation and migration and enhanced the sensitivity of cancer cells to anti-tumor drugs	[20]
Cancer cell membrane-derived nanocarrier (mCas9-sGNRs)	MDA-MB-231-tumor-bearing mice model	BIRC5 gene	Adenocarcinoma of breast	intravenous injection	/	every 3 days for 18 days	enhanced antitumor efficacy and reduced tumor thermal tolerance	[43]
Macrophage membrane-coated polyplexes (PD/P@M)	hepatic ischemia-reperfusion injury (IRI) in mice	Alox12	hepatic ischemia-reperfusion injury (IRI)	tail-vein injection	plasmids (30 µg)	day 1, day 3, and day 5 for 3 times	the level of aspartate aminotransferase (AST) and alanine aminotransferase (ALT) reduced tumor necrosis factor-α (TNF-α) and interferon-gamma (IFN-γ) level	[80]
	concanavalin-A (ConA) induced hepatic fibrosis in mice	Fas gene	hepatic fibrosis	tail-vein injection	plasmids (30 µg)	weekly for 4 weeks	validated the amelioration of the hepatic inflammation and fibrosis	
	concanavalin-A (ConA) induced fulminant hepatic failure in mice	Fas gene	fulminant hepatic failure	tail-vein injection	plasmids (30 µg)	day 1, day 3, and day 5 for 3 times	reduced hyperemia and prolonged the survival time of the model mice	
Carboxylated branched poly (β-amino ester)	mouse glioma model	ReNL reporter gene	murine glioma tumors	intracranial injection	10 µL with RNP (15 *w*/*w*%)	single	bright ReNL fluorescence within the tumor bulk, the brightest ReNL signal was localized in closest proximity to the injection site	[52]
Core–shell hollow mesoporous organosilica nanoparticles	H22 tumor-bearing mice model	EGFR gene	hepatocellular carcinoma (HCC)	intravenous injection	plasmids (2.5 mg/kg)	every 2 days for 22 days	achieved efficient EGFR gene therapy and caused 85% tumor inhibition in a mouse model, showed high accumulation at the tumor site in vivo, and exhibited good safety with no damage to major organs	[45]
P/M@CasMTH1	A549 tumor-bearing Balb/c nude mice	MTH1 gene	tumor	tail-vein injection	20 mg/kg	single	destroyed the self-defense system of tumor cells and led to the inhibition of tumor progression in vivo	[57]
Gold nanoclusters (GNCs)	WT-C57BL/6 mice	Pcsk9 gene	cardiovascular Diseases	intravenous injection	/	single	achieved efficient gene editing of Pcsk9 in the liver and the down-regulation of serum LDL-C	[29]
Gold nanorods (AuNRs)	ConA-Induced Hepatic Fibrosis	Fas gene	chronic hepatitis	intravenous injection	plasmids (2.0 μg/kg)	for 4 weeks after 48 h of each injection of ConA	delivered CMV-Cas9-Fas to the hepatocytes and rescued the mice from chronic hepatitis	[26]
MG-RNP@CaCO_3_	A549 tumor-bearing BALB/c nude mice	Nrf2 gene	tumor	intravenous injection	MG-RNP@CaCO_3_ (16 mg/kg)	every 3 days for 7 days	reduced the Nrf2 expression and suppressed tumor growth	[56]
Glutathione (GSH)-responsive silica nanocapsules (SNCs)	wild-type mice	App- and Th- gene	central nervous system (CNS) disorders	intravenous injection	total RNA (5 mg/kg)	every 5 days for 3 times	led to 19.1% reduction in the expression level of intact APP and 30.3% reduction in the expression level of TH	[35]
Angiopep-2-functionalized, disulfide-cross-linked nanocapsules	GBM tumor-bearing mice	PLK1 gene	glioblastoma	intravenous injection	RNP (1.5 mg/kg)	single	marked inhibition of GBM tumor growth and the approximate trebling of median survival time	[54]
Angiopep-2 decorated, guanidinium and fluorine functionalized polymeric nanoparticle	orthotopic U87MG-Luc glioma-bearing nude mice	PLK1 gene	glioblastoma	intravenous injection	Cas9 (15 μ g)	every 2 days for 5 times	suppressed tumor growth and improved the median survival time of mice bearing orthotopic glioblastoma to 40 days	[53]
Black phosphorus nanosheets (BPs)	A549/EGFP tumor-bearing nude mice model	EGFP	tumor	intra-tumoral injection	50 μL of Cas9N3-BPs (100 μg/mL BPs and 800 nM Cas9N3 in PBS	single	reduced the EGFP signals around the site of injection	[81]
Guanidium-rich lipopeptide GD-LP	Duchenne muscular dystrophy (DMD) mouse model	DMD gene	Duchenne muscular dystrophy (DMD)	intramuscular injection	20 µL (0.6 μg/μL RNP and 0.18 μg/μL ssODN)	single	restored dystrophin expression, reduced skeletal muscle fibrosis, and significantly improved muscle strength	[51]
pCas9-loaded nanocore (PRTM/pCas9/Ca; NP)	H1299-Luc xenograft model	HIF-1α	tumor	intravenous injection	plasmids (10 ug)	single	augmented the therapeutic efficacy of PTX, causing distinct apoptosis and noticeable tumor suppression	[61]
P-aP-DTT-LMP-g4 polyplex NP	C666-1 xenograft tumor model	Lmp1 gene	nasopharyngeal carcinoma (NPC)	peritumoral injection	plasmids (10 mg)	on days 1, 4, 7, and 10	achieved good tumor penetration and tumor growth inhibition	[82]
Hyperbranched polyamide amine (HPAA)	HNE-1 cells-bearing mice	SGK3 gene	nasopharyngeal carcinoma (NPC)	intravenous injection	/	every 2 days for 21 days	inhibited angiogenesis and tumor cell proliferation	[50]
Methoxy-poly(ethyleneglycol)-b-poly(2-(azepan-1-yl) ethyl methacrylate) (mPEG-PC7A)	Mdx Mice	DMD gene	Duchenne muscular dystrophy	intramuscular injection	1.2 µg/µL RNP and 0.36 µg/µL ssODN	single	exhibited less muscle fibrosis and fewer pathological characteristics of muscular dystrophy, as well as interstitial fibrosis	[59]
Programmable unlocking of the nano-matryoshka-CRISPR system (PUN)	B16-F10 xenograft tumor model.	PD-L1 and PTPN2 gene	melanoma	tail-vein injection	plasmids (5 µg)	every 2 days	activated cascade amplified adaptive immunity and induced long-term immune memory effect.	[83]
Au nanoparticle-loaded core–shell tecto dendrimers (Au CSTDs)	B16F10 tumor-bearing mice	PD-L1 gene	melanoma	intra-tumoral injection	Cas9-PD-L1 (10 μg)	every 3 days for 12 days	increased the distribution of CD4+/CD8+ T cells, reduced the proportion of immunosuppressive cells, and upregulated the cytokines TNF-α/IFN-γ/IL-6	[24]
F-PC/pHCP	B16F10 tumor-bearing mice	PD-L1 gene	melanoma	peritumoral injection	plasmids (1 mg/kg)	single	potentiated immunotherapy efficacy through a combination of PD-1/PD-L1 checkpoint blockade and tumor immune microenvironment reprogramming and triggered an immune memory response to increase the proportions of Tem and Tcm cells in the spleen, achieving efficient suppression of distant tumor growth and lung metastasis	[64]
Fe_3_O_4_ @mPDA-mPEG-Ni	B16F10 tumor-bearing mice	PD-L1 gene	tumor	tail-vein injection	/	single	the mild photothermal stimulated anti-tumor immune responses without causing damage to normal tissues like skin, and also could promote the specific release of RNP in tumors, and the gene knock-out efficiency on the PD-L1 gene in melanoma cells in vivo was about 25.1%,	[36]
UCNPs-Cas9@PEI	A549-tumor-bearing mice	PKL1	tumor	intratumor injection	100 uL, 3.5 mg/mL	every 3 days for 20 days	targeted PKL-1gene and inhibited the proliferation of tumor cell	[84]
UCNPs-Cas9@CM	HBV-Tg mice	HBV	chronic hepatitis B virus (HBV) infection	tail-vein injection	RNP (40 μg)	every day for 14 days	decreased serum levels of HBV DNA, HBsAg and HBeAg, as well as the HBsAg and HBcAg levels in hepatocytes	[55]
CuS-RNP/DOX@PEI	A375-tumor-bearing BALB/c mice	Hsp90	tumor	intra-tumoral injection	nanocomposites (5 mM)	every 3 days for 3 weeks	potentially tumor synergistic therapy, including GT, mild-PTT, and CT. Photothermal controlled gene editing to disrupt Hsp90α provides a potential strategy to reduce tumor thermal tolerance for enhanced mild-PTT effects.	[85]
ZIF8-NaHCO_3_@Cas9 (ZNC)	OVX-induced osteoporosis mice model	nuclear factor kappa-B ligand (RANKL)	osteoporosis	femoral marrow cavity injection	/	single	released the carried NaHCO_3_ to achieve acid neutralization and reduce ROS level, inhibiting RANKL expression and osteoclast activity and suppressing the expression of RANKL, reducing the formation of osteoclasts and effectively cutting off the source of acidic micro-environment formation.	[46]
MG-RNP@CaCO_3_	A549 tumor-bearing BALB/c nude mice	Nrf2 gene	/	intravenous injection	16 mg/kg	every 3 days for 7 times	inhibited tumor growth and protected normal surrounding tissue from oxidative stress.	[56]
CuS-RNP@PEI	B16F10 tumor-bearing orthotopic mouse model	PTPN2	malignant neoplasm	peritumor injection	/	every 2 days for 3 times	tumor tissues with indicated treatments and with no noticeable abnormality nor appreciable organ damage	[86]
Multistage-sensitive nanocomplex (MUSE)	B16F11 tumor-bearing mice model	CD47, PD-L1	tumor	intravenous injection	0.25 mg/kg	once every 3 days for 8 times	activated robust CD8+ T cells and M1 macrophage-mediated adaptations and root ions anti-tumor immune response and triggered long-lasting immune memory action, led to significant inhibition of tumor growth and improved survival with virtually undetectable off-target delivery effects	[60]
LNP	cystic fibrosis (CF) mouse model	CFTR	cystic fibrosis	tail-vein injection	2 mg/kg	once a week for 3 times	2.34% of CFTR gene extracted from whole lung tissue was corrected, corrected G542X mutation in mouse lungs	[87]
LNP	ΔEx44 DMD mice model	DMD gene	Duchenne muscular dystrophy (DMD)	tail-vein injection	sg DMD (1 mg/kg)	once a week for 3 times	4.2% of dystrophin protein in TA muscles	[88]
LNP	OV8-bearing mice	PLK1 gene	ovarian tumors	intraperitoneal injection	0.75 mg/kg	single	targeted treatment of disseminated tumors and increased overall survival by ~80%	[71]
	005 GBM-bearing mice	PLK1 gene	glioblastoma	intracerebral injection	0.05 mg/kg	single	reduced tumor growth and increased median survival from 32.5 to >48 days	
LNP-INT01	/	transthyretin (Ttr) gene	TTR amyloidosis (ATTR)	tail-vein injection	1 mg/kg	single	editing in the liver reached nearly ~70%, while serum levels of TTR were reduced more than 90%	[16]
LNP-NTLA-2001	patients with hereditary ATTR amyloidosis with polyneuropathy	transthyretin (Ttr) gene	TTR amyloidosis (ATTR)	intravenous injection	total RNA (0.1 mg/kg or 0.3 mg/kg)	within an ongoing phase 1 clinical study	led to a decrease in serum TTR protein concentrations with only mild adverse events	[89]
LNP (ZAPL75C)	MDA-MB-468 cells tumor-bearing mice	FOXC1 gene	tumor	intravenous injection	plasmids (20 μg)	two injections	∼42.2% and ∼82.2% reduction in tumor volume	[72]
Peptide-conjugated lipids	SAS/luc-bearing mouse model	HuR gene	tumor	intravenous injection	10 mg/kg	single	targeted to tumor sites and induced apoptosis in more tumor cells	[73]
LNP	hemophilia A and B mouse model	SERPINC1 gene	hemophilia A and B	intracerebral injection	1.2 mg/kg	three times with 2-week	reduced spontaneous bleeding and secondary hemophilia complications by enhanced thrombosis potential	[15]
Nano-cleaver (HepCCCleaver)	HBV replication mouse models	HBV	hepatitis B virus infection	intravenous injection	20 mg/kg	single	resulted in decreased levels of HBsAg by 48.6%, HBeAg by 58.7%, HBV DNA by 53.5%, and HBV RNA by 56.3% and achieved efficient virus elimination to treat HBV infection in vivo	[75]
LNP	DMD exon 45 knock-in (hEx45KI) mice	DMD gene	Duchenne muscular dystrophy (DMD)	intramuscular injection	mRNA (10 μg)	single	generated a mouse model of DMD	[76]
LNP (P127 M@pCD98)	IL-10 knock-out C57/BL6 mice	CD98	Chronic Colitis	oral injection	plasmids (1 μg)	every other day for 15 days	decreased CD98 expression, down-regulated pro-inflammatory cytokines (TNF-***α*** and IL-6), up-regulated anti-inflammatory factors (IL-10), and polarized macrophages to M2 phenotype	[68]
LNP (iLP181)	HepG2-Luc tumor-bearing mice	PLK1	tumor	intra-tumoral injection	plasmids (0.5 mg/kg)	every 2–4 days for 21 days	achieved tumor growth suppression by gene abolishment of PLK1	[70]
4A3-SC8 dLNPs	HEK293 B/GFP tumor-bearing mice	GFP	tumor	intra-tumoral injection	0.5 mg/kg.	single	>20% HDR-mediated gene correction in vivo	[67]

**Table 3 nanomaterials-15-00540-t003:** The chemical structures of the components for organ-targeted LNP.

Represents Liposomes	Target Organ	Formulation (Molar Ratio)	Refs.
9A1P9-5A2-SC8	liver	9A1P9:5A2-SC8:cholesterol:DMG-PEG2000 = 25:30:30:1	[122]
9A1P9-DDAB iPLNPs	lung	9A1P9:DDAB:cholesterol:DMG-PEG2000 = 60:30:40:0.4	
10A1P16-MDOA iPLNPs	spleen	10A1P16:MDOA:cholesterol:DMG-PEG2000 = 25:30:30:1	
18:1 DAP (DODAP)	liver	5A2-SC8:DOPE: cholesterol:C14PEG:DODAP = 19:19:38:4:20	[119]
18:1 TAP (DOTAP)	lung	5A2-SC8:DOPE:cholesterol:C14PEG:DOTAP = 11.9:11.9:23.8:2.4:50	
18:1 PA (18PA)	spleen	5A2-SC8:DOPE:cholesterol:C14PEG:18PA = 16.7:16.7:33.3:3.3:30	
306-O12B LNP	liver	306-O12B:cholesterol:DOPC:DMG-PEG = 50:38.5:10:1.5	[123]
DOTAP40	lung	5A2-SC8:DOPE:cholesterol:DMG-PEG = 24:24:47:5	[87]
mDLNP-2	liver	5A2-SC8:DOPE:cholesterol:DMG-PEG = 36:20:40:4	
20% DODAP 4A3-SC8	lung	4A3-SC8:DOPE:cholesterol:DMG-PEG:DODAP = 15:15:30:3:16	[118]
50% DOTAP 4A3-SC8	liver	4A3-SC8:DOPE: cholesterol:DMG-PEG:DOTAP = 15:15:30:3:63	
10% 18PA 4A3-SC8	spleen	4A3-SC8:DOPE:cholesterol:DMG-PEG:18PA = 15:15:30:3:7	
FX12m	liver	BP lipid 312:DOPE: cholesterol:DMG-PEG-2000:cholesterol-PEG-2000 = 46.0:12.4:40.0:1.2:0.4	[120]
FC8m	lung	ADC:Lipid III-45:DOPE:cholesterol:DMG-PEG-2000 = 46.0:24.0:12.5:16.0:1.5	
LNP-INT01	liver	LP01 lipid:cholesterol:DSPC:PEG2000-DMG = 45:44:9:2	[16]
RCB-4-8 LNPs	lung	ionizable lipids (RCB):DOTAP:cholesterol:C14-PEG2000 = 30:39:30:1	[69]

## Data Availability

No new data were created or analyzed in this study.

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
