# Peer review of "Precisely Targeted Nanoparticles for CRISPR-Cas9 Delivery in Clinical Applications"

_nanomaterials, 2025, doi:10.3390/nano15070540_

Round 1
Reviewer 1 Report
Comments and Suggestions for Authors
Overall Assessment:
This manuscript provides a comprehensive and well-structured review of chemically synthesized nanoparticles (NPs) for CRISPR/Cas9 delivery. The authors discuss key considerations in nanoparticle design, encapsulation strategies, targeting mechanisms, and clinical applications. The review is well-referenced and covers a broad range of recent advancements.
However, several aspects need improvement before the manuscript can be considered for publication. These include enhancing critical discussion, improving clarity and organization, ensuring consistency in referencing, addressing figure-related issues, and incorporating more illustrative examples.
Below are specific points for improvement.
Major points for improvement:
- Need for more critical analysis
While the manuscript provides an extensive list of studies on nanoparticle-based CRISPR/Cas9 delivery, it lacks a strong critical analysis of the advantages and limitations of different approaches. The following areas should be expanded:
- A comparative evaluation of different nanoparticle systems' effectiveness, biocompatibility, and stability.
- Discuss safety concerns, immune responses, and toxicity associated with various delivery vectors.
- An overview of challenges in clinical translation, including regulatory hurdles and manufacturing scalability.
Currently, the manuscript leans heavily on describing existing studies without a deep discussion of which strategies are most promising and why. A dedicated section on challenges and future directions would strengthen the impact of the review.
- Organization and flow
The manuscript would benefit from more explicit section structuring to improve readability.
- Some topics, such as encapsulation strategies, are discussed in multiple sections, leading to redundancy.
- The introduction should provide a more precise roadmap for the review.
- Consider reordering sections to discuss general nanoparticle design principles, then move to specific strategies for CRISPR/Cas9 delivery, followed by applications and clinical prospects.
- Clarity and readability
Specific sentences are overly complex and should be revised for clarity. Example:
- Original: "Nanotechnology has been widely investigated in the fields of gene therapeutic agent delivery such as antisense oligonucleotides, siRNAs, microRNAs, mRNAs, and the CRISPR/Cas9 system."
- Suggested revision: "Nanotechnology is widely used for delivering gene therapy agents, including antisense oligonucleotides, siRNAs, microRNAs, mRNAs, and CRISPR/Cas9."
Additionally:
- Some terms (e.g., "plasmids,” "RNP,” "lipid nanoparticles") should be consistently defined upon first use.
- The explanations of technical aspects (such as proton sponge effects, base-pair interactions) should be simplified for broader accessibility.
- Figure 1 is illegible
- The resolution of Figure 1 is too low, making the text and details challenging to interpret.
- Suggested improvements:
- Use a higher-resolution version of the figure.
- If the figure contains small text, increase font size and contrast.
- If too much information is packed into one figure, consider splitting it into sub-figures (e.g., Figure 1A, 1B, etc.).
- Provide a detailed caption explaining the key elements.
- Lack of other illustrative examples
- The manuscript contains few figures or graphical representations. Additional diagrams would help clarify complex concepts.
- Consider adding:
- A schematic of different nanoparticle types used for CRISPR/Cas9 delivery.
- A flowchart comparing viral vs. non-viral delivery methods.
- Illustrations showing targeting mechanisms (e.g., ligand-receptor interactions, nanoparticle uptake pathways).
- Figures would enhance readability and engagement for the audience.
- Inconsistent referencing and citation formatting
- Some references appear out of order or are repeated.
- Ensure uniform citation formatting (e.g., some references use full journal names, others abbreviate them).
- Verify that all cited studies are referenced correctly in the bibliography.
- More emphasis on clinical translation and regulatory considerations
- While the manuscript discusses the potential of CRISPR/Cas9 nanoparticles, it does not address regulatory challenges.
- Suggested additions:
- Current status of FDA/EMA regulations for gene-editing therapies.
- Limitations in large-scale manufacturing of CRISPR-loaded nanoparticles.
- Potential ethical concerns related to in vivo genome editing.
A brief discussion of ongoing clinical trials using nanoparticle-based CRISPR/Cas9 delivery would also strengthen the review.
- More emphasis on future directions
- The conclusion should be expanded to include:
- Emerging technologies (e.g., AI-driven nanoparticle design, bioengineered exosome delivery).
- Potential breakthroughs in nanoparticle chemistry to improve biocompatibility.
- Unanswered questions and research gaps in the field.
The manuscript summarizes past research well but does not provide enough forward-looking insights.
Minor points for improvement:
- Grammar and typographical errors:
- Example: "so that provided a patch of local negative charges" → "which provided a patch of local negative charges."
- Standardizing terminology:
- Ensure consistency in how nanoparticle types and gene-editing terms are referred to (e.g., "Cas9 RNP" vs. "CRISPR/Cas9 RNP").
- Figures and diagrams:
- Consider including a visual summary table comparing different nanoparticle types.
Recommendation:
The manuscript presents an important and timely review of nanoparticle-based CRISPR/Cas9 delivery. However, major revisions are required to improve clarity, organization, figure quality, critical analysis, and discussion of clinical translation.
Comments on the Quality of English Language
English needs a native speaker review and edits.
Author Response
Response to reviewers’comments
Dear editors and reviewers:
On behalf of all the contributing authors, I would like to express our sincere appreciations of your letter and reviewers’ professional review work and constructive comments on our review entitled “Precisely targeted nanoparticles for CRISPR/Cas9 delivery in clinical applications” (Manuscript ID: nanomaterials-3494201). These comments are all valuable and helpful for improving our article. According to the associate editor and reviewers’ comments, we have made extensive modifications to our manuscript and supplemented extra data to make our review convincing.
As you are concerned, there are several problems that need to be addressed. In this revised version, according to your nice suggestions, we have made extensive corrections to our previous draft, the detailed corrections are listed below. The reviewers’ comments are laid out below in italicized font and specific concerns have been numbered. Our response is given in normal font and changes/additions to the manuscript are given in the red text.
We understand that your valuable feedback will help us further improve the quality and clarity of our work. If you have any specific questions or need additional information, please do not hesitate to contact us at baoji@scu.cn.
Thank you again for your dedication and contribution to the advancement of knowledge in this field.
Best regards,
Ji Bao
Review1#
- Need for more critical analysis
While the manuscript provides an extensive list of studies on nanoparticle-based CRISPR/Cas9 delivery, it lacks a strong critical analysis of the advantages and limitations of different approaches.
The following areas should be expanded:
- A comparative evaluation of different nanoparticle systems' effectiveness, biocompatibility,
and stability.
- Discuss safety concerns, immune responses, and toxicity associated with various delivery
vectors.
- An overview of challenges in clinical translation, including regulatory hurdles and
manufacturing scalability.
Currently, the manuscript leans heavily on describing existing studies without a deep discussion of which strategies are most promising and why. A dedicated section on challenges and future
directions would strengthen the impact of the review.
Response 1: We have supplemented the discussion of these issues in section 6.
- Organization and flow
The manuscript would benefit from more explicit section structuring to improve readability.
- Some topics, such as encapsulation strategies, are discussed in multiple sections, leading to redundancy.
- The introduction should provide a more precise roadmap for the review.
- Consider reordering sections to discuss general nanoparticle design principles, then move to specific strategies for CRISPR/Cas9 delivery, followed by applications and clinical prospects.
Response 2: 1. We did find some duplication of discussion that has been removed from the manuscript.
- We rewrote the introduction according to the structure of the article, and we have submitted a graphical abstract to facilitate readers' more intuitive understanding of the framework and main content of the review.
- Clarity and readability
Specific sentences are overly complex and should be revised for clarity. Example:
- Original: "Nanotechnology has been widely investigated in the fields of gene therapeutic agent delivery such as antisense oligonucleotides, siRNAs, microRNAs, mRNAs, and the CRISPR/Cas9 system."
- Suggested revision: "Nanotechnology is widely used for delivering gene therapy agents, including antisense oligonucleotides, siRNAs, microRNAs, mRNAs, and CRISPR/Cas9."
Additionally:
- Some terms (e.g., “plasmids,” “RNP,” “lipid nanoparticles”) should be consistently defined upon first use.
- The explanations of technical aspects (such as proton sponge effects, base-pair interactions) should be simplified for broader accessibility.
Response 3: 1. We have simplified this sentence in the revised manuscript based on your suggestion.
- Thanks for your careful checks. We are sorry for our carelessness. Based on your comments, we have made the corrections to make the word harmonized within the whole manuscript.
- Maybe the explanations should be simplified is protein-ligand interactions, we have revised to simplify the interpretation of these sections.
- Figure 1 is illegible
- The resolution of Figure 1 is too low, making the text and details challenging to interpret.
- Suggested improvements:
Use a higher-resolution version of the figure.
If the figure contains small text, increase font size and contrast.
If too much information is packed into one figure, consider splitting it into sub-figures (e.g., Figure 1A, 1B, etc.).
Provide a detailed caption explaining the key elements.
Response4: We have revised the figure1 to enhance readability, and we appreciate the valuable suggestion.
- Lack of other illustrative examples
- The manuscript contains few figures or graphical representations. Additional diagrams would help clarify complex concepts.
- Consider adding:
A schematic of different nanoparticle types used for CRISPR/Cas9 delivery.
A flowchart comparing viral vs. non-viral delivery methods.
Illustrations showing targeting mechanisms (e.g., ligand-receptor interactions, nanoparticle uptake pathways).
- Figures would enhance readability and engagement for the audience.
Response4: 1. We took the reviewer's suggestion and added a schematic of different nanoparticle types used for CRISPR-Cas9 delivery to the revised manuscript.
- Our manuscript is primarily a review of the use of non-viral vectors in the delivery of CRISPR/Cas9, and there are many similar flowcharts for the comparison of viral vectors and non-viral vectors in more systematic reviews, so we will not repeat it.
- We have provided a graphical abstract for the editors with clarification on showing targeting mechanisms (e.g., ligand-receptor interactions, nanoparticle uptake pathways).
- Inconsistent referencing and citation formatting
- Some references appear out of order or are repeated.
- Ensure uniform citation formatting (e.g., some references use full journal names, others abbreviate them).
- Verify that all cited studies are referenced correctly in the bibliography.
Response6: Thanks for your careful checks. We are sorry for our carelessness. Based on your comments, we have made the corrections to make the uniform citation formatting within the whole manuscript, and removed duplicate references and corrected the order of references.
- More emphasis on clinical translation and regulatory considerations
- While the manuscript discusses the potential of CRISPR/Cas9 nanoparticles, it does not
address regulatory challenges.
- Suggested additions:
Current status of FDA/EMA regulations for gene-editing therapies.
Limitations in large-scale manufacturing of CRISPR-loaded nanoparticles.
Potential ethical concerns related to in vivo genome editing.
A brief discussion of ongoing clinical trials using nanoparticle-based CRISPR/Cas9 delivery would also strengthen the review.
Response 7: We sincerely appreciate the valuable comments. We checked more relevant literature carefully and added more references on and into the Challenges and Future Directions part in the revised manuscript.
- More emphasis on future directions
- The conclusion should be expanded to include:
Emerging technologies (e.g., AI-driven nanoparticle design, bioengineered exosome
delivery).
Potential breakthroughs in nanoparticle chemistry to improve biocompatibility.
Unanswered questions and research gaps in the field.
Response 8: Thanks to the reviewers for their professional and constructive suggestions, we have added the discussion in Chapter 6.
Reviewer 2 Report
Comments and Suggestions for Authors
I reviewed the manuscript Precisely targeted nanoparticles for CRISPR/Cas9 delivery in clinical applications by Xinmei Liu et al.
Overall, I think that the manuscript is suitable for publication. Cas9 delivery by NPs is a very challenging theme in the literature and I appreciate the authors' work.
I have just a couple comments:
- first, please make sure that Figure 1 can be increased in size, as the name of the lipids are too small as they are displayed now, maybe you can just increase the font size in the description.
- I noticed that the targeted delivery of Cas9 systems is basically limited to clearance organs, i.e. liver, spleen, and kidneys. I suggest authors address this huge limitation of currently published nanosystems, either in the dedicated section (targeting) or in the conclusions.
Author Response
Response to reviewers’comments
Dear editors and reviewers:
On behalf of all the contributing authors, I would like to express our sincere appreciations of your letter and reviewers’ professional review work and constructive comments on our review entitled “Precisely targeted nanoparticles for CRISPR/Cas9 delivery in clinical applications” (Manuscript ID: nanomaterials-3494201). These comments are all valuable and helpful for improving our article. According to the associate editor and reviewers’ comments, we have made extensive modifications to our manuscript and supplemented extra data to make our review convincing.
As you are concerned, there are several problems that need to be addressed. In this revised version, according to your nice suggestions, we have made extensive corrections to our previous draft, the detailed corrections are listed below. The reviewers’ comments are laid out below in italicized font and specific concerns have been numbered. Our response is given in normal font and changes/additions to the manuscript are given in the red text.
We understand that your valuable feedback will help us further improve the quality and clarity of our work. If you have any specific questions or need additional information, please do not hesitate to contact us at baoji@scu.cn.
Thank you again for your dedication and contribution to the advancement of knowledge in this field.
Reviewer 2#
- first, please make sure that Figure 1 can be increased in size, as the name of the lipids are too small as they are displayed now, maybe you can just increase the font size in the description.
Response1: We have revised the figure1 to enhance readability, and we appreciate the valuable suggestion.
- I noticed that the targeted delivery of Cas9 systems is basically limited to clearance organs, i.e. liver, spleen, and kidneys. I suggest authors address this huge limitation of currently published nano systems, either in the dedicated section (targeting) or in the conclusions.
Response 2: Thanks for the suggestions, and we reviewed the relevant literature and supplemented possible resolution strategies in Section 3.3.
Best regards,
Ji Bao
Reviewer 3 Report
Comments and Suggestions for Authors
The review article by Liu et al. provides a comprehensive and timely discussion on nanoparticle-based CRISPR-Cas9 delivery. Given the growing interest in precise and efficient genome editing, this review addresses an important topic that will be valuable to researchers in the field. I support its publication after the following concerns are addressed.
- Nanoparticle-Based Delivery Platforms:
The review briefly mentions different nanoparticle-based delivery strategies for CRISPR-Cas9, but it would benefit from a more detailed discussion of the three main delivery platforms: plasmid DNA, mRNA, and ribonucleoproteins (RNPs). Each platform has distinct advantages and limitations in terms of efficiency, stability, and immunogenicity. Providing a structured comparison would enhance clarity and completeness.
- Standardized Notation of CRISPR-Cas9:
The term CRISPR/Cas9 appears throughout the manuscript, but the more widely accepted and standard notation in current literature is CRISPR-Cas9 (with a hyphen). Standardizing this terminology would improve clarity and alignment with contemporary publications.
- Clarification of CRISPR-Cas9 Definition:
In both the Abstract and Introduction, CRISPR-Cas9 should be correctly introduced as Clustered Regularly Interspaced Short Palindromic Repeats / CRISPR-associated protein 9.
- Minor Corrections:
・In Section 2.1.3, "cas9" should be corrected to "Cas9", following standard capitalization conventions.
・In Section 2.1.4, the term N/P ratio is used without explanation. Since this ratio is critical in nanoparticle formulation, providing a brief definition for a broader audience would improve accessibility.
- Standardized Terminology for Lipofectamine 2000:
In Sections 2.2.1 and 4.7, "Lipo-2000" is used, but this is not the standard notation for Lipofectamine 2000. To maintain consistency and clarity, it should be referred to as Lipofectamine 2000 throughout the manuscript.
Author Response
Response to reviewers’comments
Dear editors and reviewers:
On behalf of all the contributing authors, I would like to express our sincere appreciations of your letter and reviewers’ professional review work and constructive comments on our review entitled “Precisely targeted nanoparticles for CRISPR/Cas9 delivery in clinical applications” (Manuscript ID: nanomaterials-3494201). These comments are all valuable and helpful for improving our article. According to the associate editor and reviewers’ comments, we have made extensive modifications to our manuscript and supplemented extra data to make our review convincing.
As you are concerned, there are several problems that need to be addressed. In this revised version, according to your nice suggestions, we have made extensive corrections to our previous draft, the detailed corrections are listed below. The reviewers’ comments are laid out below in italicized font and specific concerns have been numbered. Our response is given in normal font and changes/additions to the manuscript are given in the red text.
We understand that your valuable feedback will help us to further improve the quality and clarity of our work. If you have any specific questions or need more information, please feel free to contact us at baoji@scu.cn.
Thank you again for your dedication and contribution to the advancement of knowledge in this field.
Reviewer 3 #
- Nanoparticle-Based Delivery Platforms:
The review briefly mentions different nanoparticle-based delivery strategies for CRISPR-Cas9, but it would benefit from a more detailed discussion of the three main delivery platforms: plasmid DNA, mRNA, and ribonucleoproteins (RNPs). Each platform has distinct advantages and limitations in terms of efficiency, stability, and immunogenicity. Providing a structured comparison would enhance clarity and completeness.
Response 1: There have been a number of reviews that have outlined this, and we have added relevant citations in the introduction section
- Standardized Notation of CRISPR-Cas9:
The term CRISPR/Cas9 appears throughout the manuscript, but the more widely accepted and standard notation in current literature is CRISPR-Cas9 (with a hyphen). Standardizing this terminology would improve clarity and alignment with contemporary publications.
Response 2: Thanks to the reviewers' professional comments, we determined the standard notation and revised in the manuscript.
- Clarification of CRISPR-Cas9 Definition:
In both the Abstract and Introduction, CRISPR-Cas9 should be correctly introduced as Clustered Regularly Interspaced Short Palindromic Repeats / CRISPR-associated protein 9.
Response 3: Thanks to the reviewers' professional comments, we revised in the manuscript.
- Minor Corrections:
・In Section 2.1.3, "cas9" should be corrected to "Cas9", following standard capitalization conventions.
・In Section 2.1.4, the term N/P ratio is used without explanation. Since this ratio is critical in nanoparticle formulation, providing a brief definition for a broader audience would improve accessibility.
Response 4: 1. Thanks for your careful checks. We are sorry for our carelessness. Based on your comments, we have made the corrections to make the word harmonized within the whole manuscript.
- We have checked the literature carefully and added more explanation of N/P ratio into the part in the revised manuscript.
-
Standardized terminology for Lipofectamine 2000:
In Sections 2.2.1 and 4.7, “Lipo-2000” is used, but this is not a standardized representation of Lipofectamine 2000. For consistency and clarity, it should be referred to as Lipofectamine 2000 throughout the manuscript.
Response 5: We sincerely thank the reviewers for their careful reading. As suggested by the reviewers, we have corrected “Lipo-2000” to “Lipofectamine 2000”.
Kind Regards,
Bao Ji
Round 2
Reviewer 1 Report
Comments and Suggestions for Authors
The revised manuscript presents an updated and comprehensive review of chemically synthesized nanoparticles (NPs) for CRISPR/Cas9 delivery. The authors have tried to address some of the concerns raised in the first review, including improving the organization, enhancing discussion on clinical translation, and refining figures.
However, several issues persist, and additional refinements are necessary before the manuscript can be considered for publication.
________________________________________
Major Points for Improvement:
The manuscript now includes a comparative discussion of different nanoparticle systems, but the analysis remains superficial. The authors should deepen the discussion by clearly outlining the strengths and weaknesses of each system in terms of efficiency, biocompatibility, stability, and toxicity.
While there is now a brief mention of immune responses, a more thorough discussion of nanoparticle-associated toxicity and immunogenicity should be included, with references to relevant in vivo studies.
Clinical translation barriers are addressed, but the discussion lacks concrete examples from ongoing clinical trials or regulatory hurdles existing CRISPR/Cas9 therapies encounter.
The manuscript's structure has improved, particularly with a more straightforward introduction. However, some sections still contain redundancies, especially in discussions on encapsulation strategies and targeting mechanisms. These should be streamlined for better readability.
A logical transition between sections is still missing. It would help if the authors clearly outlined the relationship between nanoparticle properties and their impact on CRISPR/Cas9 efficiency before diving into specific applications.
Some complex sentences have been revised for better readability, but there are still sections with dense technical jargon. For instance, explaining proton sponge effects and endosomal escape mechanisms remains challenging for a broad audience.
The authors should consider breaking down key concepts into simpler terms and using subheadings to enhance clarity.
Figure 1 has been updated to a higher resolution, which is an improvement. However, the text within the figure is still small and difficult to read. Consider breaking it into sub-figures (e.g., Figure 1A, 1B) if necessary.
There are still no schematic representations of CRISPR/Cas9 delivery pathways. A visual showing how NPs interact with cellular components would significantly enhance comprehension.
The manuscript now mentions FDA and EMA regulations, but the discussion remains surface-level. The authors should include more detail on these therapies' specific challenges in clinical translation.
Manufacturing and scalability issues are mentioned, but real-world examples of companies or research groups working on large-scale CRISPR/Cas9 nanoparticle production would add credibility.
Some effort has been made to discuss emerging trends in nanoparticle engineering, but more emphasis is needed on AI-driven nanoparticle design and synthetic biology approaches.
The section on future directions should highlight unanswered research questions and potential breakthroughs that could improve CRISPR/Cas9 delivery efficiency and safety.
________________________________________
Minor Points for Improvement:
Minor grammatical errors remain and should be corrected.
The terminology should be standardized throughout the text (e.g., "Cas9 RNP" vs. "CRISPR/Cas9 RNP").
The authors should consider including a brief concluding statement summarizing key takeaways from the review.
________________________________________
Recommendation: The manuscript has improved since the initial submission, but further revisions are required to enhance critical analysis, organization, figure clarity, and discussions on clinical translation. Addressing these issues will significantly strengthen the manuscript's impact and readability.
Comments on the Quality of English Language
Minor grammatical errors remain and should be corrected.
Author Response
Response to reviewers’ comments
Dear editors and reviewers:
On behalf of all the contributing authors, I would like to express our sincere appreciations of your letter and reviewers’ professional review work and constructive comments on our review entitled “Precisely targeted nanoparticles for CRISPR/Cas9 delivery in clinical applications” (Manuscript ID: nanomaterials-3494201). These comments are all valuable and helpful for improving our article. According to the associate editor and reviewers’ comments, we have made extensive modifications to our manuscript and supplemented extra data to make our review convincing.
As you are concerned, there are several problems that need to be addressed again. In this revised version, according to your nice suggestions, we have made extensive corrections to our previous draft, the detailed corrections are listed below. The reviewers’ comments are laid out below in italicized font and specific concerns have been numbered. Our response is given in normal font and changes/additions to the manuscript are given in the red text. We sincerely appreciate the time and effort invested by the reviewers in evaluating our manuscript. We have tried our best to make all the revisions clear again, and we hope that the revised manuscript meets the requirements for publication. We look forward to any additional feedback or suggestions.
We understand that your valuable feedback will help us further improve the quality and clarity of our work. If you have any specific questions or need additional information, please do not hesitate to contact us at baoji@scu.cn.
Thank you again for your dedication and contribution to the advancement of knowledge in this field.
Best regards,
Ji Bao
Review1#
- The manuscript now includes a comparative discussion of different nanoparticle systems, but the analysis remains superficial. The authors should deepen the discussion by clearly outlining the strengths and weaknesses of each system in terms of efficiency, biocompatibility, stability, and toxicity.
Response 1: We sincerely thank the reviewers for highlighting the importance of a balanced evaluation of nanoparticle systems. We fully agree that biocompatibility is the foundational criterion for any delivery platform and balanced evaluation of nanoparticle systems is critical for advancing CRISPR-Cas9 therapeutics. In line with this principle, our review explicitly focused on nanoparticles that have already demonstrated acceptable safety profiles in both in vitro (assessed by cytotoxicity using CCK8/MTT assays) and in vivo models (evaluated by biocompatibility through the examination of body weight stability and the absence of hepatotoxicity in histopathological analyses), as these systems represent the most viable candidates for clinical translation. In this review, our primary focus was to highlight design strategies that directly enhance gene-editing efficiency - specifically, encapsulation efficiency, targeting precision, and controlled release - as these parameters are central to overcoming current delivery bottlenecks. Therefore, there is no deliberate emphasis on biocompatibility, stability, and toxicity assessment in this review. However, in Table 2 regarding the in - vivo application studies of nanoparticles, in the column of treatment outcomes, we have summarized the results of toxicological analyses after treatment. Moreover, we also mentioned the off - target effects and toxicity of many nanoparticles in the main text. We sincerely appreciate the reviewers' valuable feedback regarding the comparative evaluation of delivery efficiencies among nanoparticle systems, it is also a matter of great interest to us. We have expanded Table 1 to provide a comprehensive summary of (1) gene types delivered to target cells, (2) detailed editing efficiencies across different nanoparticle platforms, and (3) corresponding evaluation methodologies. To further address this important aspect, Table 2 now systematically presents in vivo editing efficiencies with additional clarification of dosage parameters used in various experimental settings. We believe these enhanced comparisons, particularly the inclusion of dosage information, will better facilitate cross-platform evaluation while maintaining experimental context. The opportunity to refine these comparisons has significantly strengthened our analysis, and we remain fully open to further clarification or additional data presentation that might assist in this critical evaluation. While our original emphasis was on efficiency-driven design, these revisions provide a more holistic framework for evaluating nanoparticle systems. We hope this strengthened analysis aligns with the reviewers’ expectations and welcome further suggestions to refine our discussion.
- While there is now a brief mention of immune responses, a more thorough discussion of nanoparticle-associated toxicity and immunogenicity should be included, with references to relevant in vivo studies.
Response 2: We appreciate the reviewer's insightful suggestion regarding the need for deeper discussion of nanoparticle-associated immunogenicity. As noted in our response to Comment #1, while our primary analytical framework emphasizes efficiency-driven design parameters, we recognize the fundamental importance of biocompatibility considerations. To address this point systematically: (1) In Table 2's "Treatment Outcomes" column, we have comprehensively cataloged all reported immunological findings from the cited in vivo studies. We hope that our response to the first comment has clarified any confusion on this issue. We greatly appreciate your creative idea and support. Thank you once again.
- Clinical translation barriers are addressed, but the discussion lacks concrete examples from ongoing clinical trials or regulatory hurdles existing CRISPR/Cas9 therapies encounter.
Response 3: We deeply appreciate the reviewer's astute observation regarding the need for concrete clinical translation perspectives. In alignment with this critical suggestion, we have systematically strengthened our discussion. After carefully reviewing the literature again, it is found that currently, there are only two cases of nanoparticle - mediated CRISPR-Cas9 for in - vivo gene editing that have entered clinical trials, and indeed both are facing numerous problems. We have supplemented relevant discussions in Section 6.1.3. For instance, in bluebird bio's clinical trials, due to cancer cases among the subjects, other similar clinical trials were forced to halt, and the company is jointly investigating with the FDA. While our original analytical focus centered on preclinical design optimization (as outlined in Response #1), we recognize the growing importance of clinical-translational awareness. Thank you again for your suggestions.
- The manuscript's structure has improved, particularly with a more straightforward introduction. However, some sections still contain redundancies, especially in discussions on encapsulation strategies and targeting mechanisms. These should be streamlined for better readability.
Response 4: The encapsulation efficiency of nanoparticles for CRISPR - Cas9 directly impacts the gene - editing efficiency. Moreover, delivering CRISPR - Cas9 to specific organs or cells while avoiding toxicity caused by unnecessary editing remains a major obstacle hindering its in - vivo application. Moreover, the targeted delivery of CRISPR - Cas9 is basically limited to clearance organs, i.e. the liver, the spleen, and the kidneys, which also poses a research challenge as we mentioned in Section 3.3. Therefore, we focus on the encapsulation strategies and targeting mechanisms of different nanoparticles, aiming to provide a theoretical reference for more precisely designing effective nanoparticles.
- A logical transition between sections is still missing. It would help if the authors clearly outlined the relationship between nanoparticle properties and their impact on CRISPR/Cas9 efficiency before diving into specific applications.
Response 5: We sincerely appreciate the reviewer’s valuable suggestion to strengthen the connection between nanoparticle properties and their functional outcomes. While our response to Comment #7 and Comment #15 focused on hierarchical structuring through headings, cross-sectional emphasis and graphical abstracts, these revisions specifically target the conceptual flow between fundamental properties and functional applications.
- Some complex sentences have been revised for better readability, but there are still sections with dense technical jargon. For instance, explaining proton sponge effects and endosomal escape mechanisms remains challenging for a broad audience.
Response 6: We sincerely appreciate the reviewer's thoughtful feedback regarding the balance between technical precision and accessibility. While we have implemented targeted revisions to enhance readability, terms like "proton sponge effect" and "endosomal escape" represent foundational concepts in nanomedicine with established lexical consensus, altering these might compromise precision.
- The authors should consider breaking down key concepts into simpler terms and using subheadings to enhance clarity.
Response 7: We deeply appreciate the reviewer's insightful suggestion to improve the readability of our manuscript. While we agree that subheadings can improve readability, however, in our review, each section already adopts a three - level heading format. Taking Section 2 as an example, we mainly focus on the strategies to enhance the delivery efficiency of CRISPR - Cas9, which mainly include encapsulation efficiency (in Section 2.1) and intracellular transport mechanisms (in Section 2.2). We categorize the strategies for enhancing encapsulation efficiency into four aspects, including electrostatic interaction (in Section2.1.1), base-pair (in Section2.1.2), protein-ligand interactions (in Section2.1.3) and other influencing factors (in Section2.1.4), introducing fourth-level headings might fragment the narrative flow. Instead, we have used a graphical abstract, which serves as a visual roadmap to enhance visual guidance. It now summarizes the article's framework, explicitly connecting nanoparticle design principles to their therapeutic applications. Furthermore, in each section we have provided section summaries to concise bullet points precede each major subsection.
- Figure 1 has been updated to a higher resolution, which is an improvement. However, the text within the figure is still small and difficult to read. Consider breaking it into sub-figures (e.g., Figure 1A, 1B) if necessary.
Response 8: We sincerely appreciate the reviewer’s meticulous attention to the visual clarity of Figure 1. To address this concern comprehensively, we have implemented the following enhancements: 1. We have increased all the font size of Figure 1 to enhance its legibility. 2.We have exported the images with higher-definition quality, simultaneously, we have furnished the editors with both PPT and PDF versions of the figure. This is to preclude any blurring of the image resulting from compression, as such an issue might impinge upon the readers' comprehension. This constructive feedback has significantly improved the pedagogical value of our graphical presentation.
9.There are still no schematic representations of CRISPR/Cas9 delivery pathways. A visual showing how NPs interact with cellular components would significantly enhance comprehension.
Response 9: We sincerely appreciate the reviewer's insightful suggestion to enhance the mechanistic clarity of our discussion. In response to this valuable feedback. We have provided a graphical abstract for the editors with clarification on showing targeting mechanisms (e.g., ligand-receptor interactions, nanoparticle uptake pathways).
- The manuscript now mentions FDA and EMA regulations, but the discussion remains surface-level. The authors should include more detail on these therapies' specific challenges in clinical translation.
Response 10: We sincerely appreciate the reviewer's expert guidance in strengthening the regulatory perspective of our analysis. After conducting a thorough review of the literature, we found that currently only two related studies have advanced to the clinical trial stage. In Section 5.1.1, we have expanded the content by conducting a dedicated analysis of the two ongoing LNP - based CRISPR trials: Intellia's NTLA - 2001 (NCT04601051) and Vertex/CRISPR Therapeutics' CTX001 (NCT03655678). We have provided detailed information on their specific regulatory milestones. Additionally, relevant discussions have been supplemented in Chapter 6. As noted in our response to Comment #1, while our primary analytical framework focuses on preclinical design optimization strategies, we recognize the growing importance of early-stage regulatory awareness. These enhancements provide targeted insights into translation barriers without diluting our core focus on delivery engineering.
11.Manufacturing and scalability issues are mentioned, but real-world examples of companies or research groups working on large-scale CRISPR/Cas9 nanoparticle production would add credibility.
Response 11: We sincerely appreciate the reviewer's valuable suggestion to strengthen the practical dimension of our scalability discussion. While our core analytical framework remains centered on design-stage nanoparticle engineering (as outlined in Response #1), these additions provide concrete anchors for translational feasibility discussions. We have carefully curated examples that directly inform preclinical design choices. Should the reviewer recommend deeper exploration of commercial-scale bioprocessing, we would be pleased to incorporate additional case studies from recent FDA Emerging Technology Program reports. To address this critical gap while maintaining methodological focus, we have implemented the following targeted enhancements in Section6.1.2 and 6.2.2.
- Some effort has been made to discuss emerging trends in nanoparticle engineering, but more emphasis is needed on AI-driven nanoparticle design and synthetic biology approaches.
Response 12: We sincerely appreciate the reviewer’s valuable suggestion to further emphasize cutting-edge methodologies in nanoparticle engineering. To address this critical insight while maintaining methodological coherence, we have implemented the following targeted enhancements. Undoubtedly, it is a field that not only merits our focused attention but also warrants our earnest efforts in learning and application within the relevant context of this study. To address the reviewer's concerns, we have significantly expanded discussions on AI-driven nanoparticle design and synthetic biology strategies, integrating cutting-edge research and concrete examples to highlight their possible transformative potential for CRISPR/Cas9 delivery.
- The section on future directions should highlight unanswered research questions and potential breakthroughs that could improve CRISPR/Cas9 delivery efficiency and safety.
Response 13: We are extremely grateful to the reviewer for this meaningful suggestion. Taking into account the previous suggestions, we have supplemented an in - depth discussion on this topic. This revision systematically addresses unresolved challenges while mapping actionable pathways for innovation, emphasizing interdisciplinary integration (AI, synthetic biology) and ethical foresight.
- The terminology should be standardized throughout the text (e.g., "Cas9 RNP" vs. "CRISPR/Cas9 RNP").
Response 14: We sincerely thank the reviewer for careful reading. We have consistently used the abbreviation RNP throughout the entire manuscript, and we have clarified the full term " ribonucleoprotein " upon its initial mention.
- The authors should consider including a brief concluding statement summarizing key takeaways from the review.
Response 15: Revised Response 15:
We sincerely appreciate the reviewer's valuable suggestion to enhance the synthesis of key concepts. In response, we have carefully revised the manuscript to:
(1) Strengthen structural clarity: Added a dedicated concluding paragraph in the Introduction that systematically outlines the review's framework, methodological focus (design strategies for CRISPR delivery optimization), and applications.
(2) Visual reinforcement: Expanded the graphical abstract to explicitly map critical nanoparticle design strategies and key points, thereby enhancing accessibility for interdisciplinary readers.
(3) Cross-sectional emphasis: Implemented progressive summarization throughout subsections (e.g., Nanoparticles deliver CRISPR-Cas9 complexes via systemic or local injection in vivo gene editing studies, and systemically administer CRISPR-Cas9 with a threshold that delivers to specific cells or tissues leading to the unwanted editing. Various approaches have been used to engineer NPs to enhance their targeting ability and therapeutic potency. Here, we reviewed the specific chemistries that have been used to develop targeted gene editing methods, including high-throughput screening platforms, cell membrane-coated nanoparticles, and receptor-ligand interaction-mediated targeting. ", in Section 3) to reinforce critical insights while maintaining narrative flow.